# Cone bipolar cell synapses generate transient versus sustained signals in parallel ON pathways of the mouse retina

Sidney P Kuo[1,2], Wan-Qing Yu[1], Prerna Srivastava[3], Haruhisa Okawa[1], Luca Della Santina[1], David M Berson[4], Gautam B Awatramani[3], Rachel OL Wong[1], Fred Rieke[1]*

[1]Department of Neurobiology and Biophysics, University of Washington, Seattle, United States; [2]Department of Biology, University of Victoria, Victoria, Canada; [3]Department of Neuroscience, University of Minnesota, Minneapolis, United States; [4]Department of Neuroscience, Brown University, Providence, United States

## eLife Assessment

This **important** study shows that retinal bipolar cell subtype-specific differences in the size of synaptic ribbon-associated vesicle pools contribute to the transient versus sustained kinetics of the responses of retinal ganglion cells. The data are extensive and **compelling**. This work will be of broad interest to researchers working on synaptic transmission, retinal signal processing, and sensory neurobiology.

*For correspondence: rieke@uw.edu

**Abstract** Parallel processing is a fundamental organizing principle in the nervous system and understanding how parallel neural circuits generate distinct outputs from common inputs is a key goal of neuroscience. In the mammalian retina, divergence of cone signals into multiple feedforward bipolar cell pathways forms the initial basis for parallel retinal circuits dedicated to specific visual functions. Here, we used patch-clamp electrophysiology, electron microscopy, and two-photon imaging of a fluorescent glutamate sensor to examine how kinetically distinct responses arise in transient versus sustained ON alpha retinal ganglion cells (ON-T and ON-S RGCs) of the mouse retina. We directly compared the visual response properties of these RGCs with their presynaptic bipolar cell partners, which we identified using 3D electron microscopy reconstruction. Different ON bipolar cell subtypes (types 5i, 6, and 7) had indistinguishable light-driven responses whereas extracellular glutamate signals around RGC dendrites and postsynaptic excitatory currents measured in ON-T and ON-S RGCs in response to the identical stimuli used to probe bipolar cells were kinetically distinct. Anatomical examination of the bipolar cell axon terminals presynaptic to ON-T and ON-S RGCs suggests that bipolar subtype-specific differences in the size of synaptic ribbon-associated vesicle pools may contribute to transient versus sustained kinetics. Our findings indicate that feedforward bipolar cell synapses are a primary point of divergence in kinetically distinct visual pathways.

## Introduction

Divergence into parallel pathways allows neural circuits to generate diverse outputs from a common set of inputs. This functional diversity can be quite striking; in vision, for example, light inputs initially encoded by the same collection of cone photoreceptors mediate sensitivity to several kinds of motion, to color and to spatial structure, all in the same region of space. These diverse functional sensitivities arise through parallel processing of light inputs, first in the retina and subsequently in downstream

visual areas. Here, we focus on the origin of transient and sustained responses – a key differentiating characteristic of retinal output cells across species (*Awatramani and Slaughter, 2000*; *Baden et al., 2016*; *Bae et al., 2018*; *Cleland et al., 1973*; *Ikeda and Wright, 1972*; *Roska and Werblin, 2001*; *Werblin, 1970*).

The retina provides a unique opportunity to study the mechanisms that differentiate signals in parallel pathways. We have a near-complete list of retinal cell types and substantial information about which of these cell types participate in specific parallel pathways (*Helmstaedter et al., 2013*; *Masland, 2012*). Under daylight conditions, retinal processing starts with cone photoreceptors transducing light inputs and providing divergent output to ~13 types of bipolar cell (*Euler et al., 2014*; *Greene et al., 2016*; *Helmstaedter et al., 2013*; *Shekhar et al., 2016*; *Tsukamoto and Omi, 2017*). The resulting bipolar cell signals are transmitted to ~20–40 retinal ganglion cell (RGC) types (*Baden et al., 2016*; *Bae et al., 2018*; *Goetz et al., 2022*). RGC responses are also strongly shaped by the activity of a large and diverse collection of amacrine cells that receive input from bipolar cells and provide presynaptic inhibition to bipolar cells and postsynaptic inhibition to RGCs (*Diamond, 2017*; *Jadzinsky and Baccus, 2013*). Finally, RGC responses are shaped by feedforward synaptic properties, particularly synaptic depression and facilitation (*DeVries, 2000*; *Grabner et al., 2016*; *Jarsky et al., 2011*; *Lagnado and Schmitz, 2015*; *Nikolaev et al., 2013*; *Oesch and Diamond, 2011*).

These mechanisms collectively produce highly distinct responses across RGC types – with cells that sample inputs on a variety of spatial and temporal scales, cells with distinct chromatic properties, cells that sense the direction of moving objects, and cells that separate local from global motion (reviewed by *Gollisch and Meister, 2010*; *Kerschensteiner, 2022*). Several mechanisms contribute to these functional differences, starting with the division of signals into ON and OFF pathways at the cone-to-bipolar synapse (*Euler et al., 2014*; *Slaughter and Miller, 1981*). Temporal properties of responses of different bipolar cells can also differ for some stimuli (*Awatramani and Slaughter, 2000*; *Baden et al., 2013*; *Euler et al., 2014*; *Euler and Masland, 2000*; *Ichinose et al., 2014*; *Ichinose and Hellmer, 2016*; *Wu et al., 2000*). In the inner retina, integration of excitatory and inhibitory synaptic inputs produces directionally selective responses in specific RGC types (*Taylor and Vaney, 2002*; *Vaney et al., 2012*) and differences in spatial weighting functions in other RGCs (*Cafaro and Rieke, 2013*; *Roska and Werblin, 2001*; *Zhang et al., 2012*). Properties of dendritic integration and RGC spike generation also differentiate responses of some RGCs (*Margolis et al., 2010*; *Ran et al., 2020*; *Wienbar and Schwartz, 2022*).

A fundamental difference among RGCs is whether their responses are sustained or transient (*Awatramani and Slaughter, 2000*; *Baden et al., 2016*; *Bae et al., 2018*; *Cleland et al., 1973*; *Ikeda and Wright, 1972*; *Roska and Werblin, 2001*). Cone photoreceptors respond to changes in light intensity with graded potentials that have both a transient and sustained component (*Angueyra and Rieke, 2013*; *Cangiano et al., 2012*; *Szikra et al., 2014*). Both kinetic components of the cone response are maintained in the firing rates of sustained cells, which track the prevailing contrast. However, only the initial component of the cone response is preserved in the spiking response of transient cells, which therefore respond primarily to changes in contrast. Where and how these differences originate is not clear. Here, we show that the distinct kinetics of responses of ON-sustained alpha and ON-transient alpha RGCs (1) persist for stimuli that elicit near-identical responses among the relevant bipolar cells, (2) persist without amacrine cell-mediated inhibition, and (3) are correlated with anatomical differences in output synapses of bipolar cell subtypes that are presynaptic to these RGCs. These findings highlight the importance of feedforward bipolar cell output synapses in producing kinetically distinct retinal outputs.

## Results

We focused on two ON RGC cell types, ON-transient alpha (ON-T) and ON-sustained alpha (ON-S) RGCs, that could be readily identified in a flat-mount preparation of the isolated mouse retina (*Figure 1A*) by their characteristic spiking response to a step increment in light intensity (*Figure 1B*; *Bae et al., 2018*; *Farrow et al., 2013*; *Goetz et al., 2022*; *Krieger et al., 2017*; *Kuo et al., 2016*). Based on their light response, dendritic field size, and stratification pattern in the inner plexiform layer (IPL) (see below; *Figure 3—figure supplement 2*), the ON-T RGCs we studied here likely correspond to the PV2 RGC type of *Farrow et al., 2013*, 6sw RGC type of *Bae et al., 2018*, and the medium-field

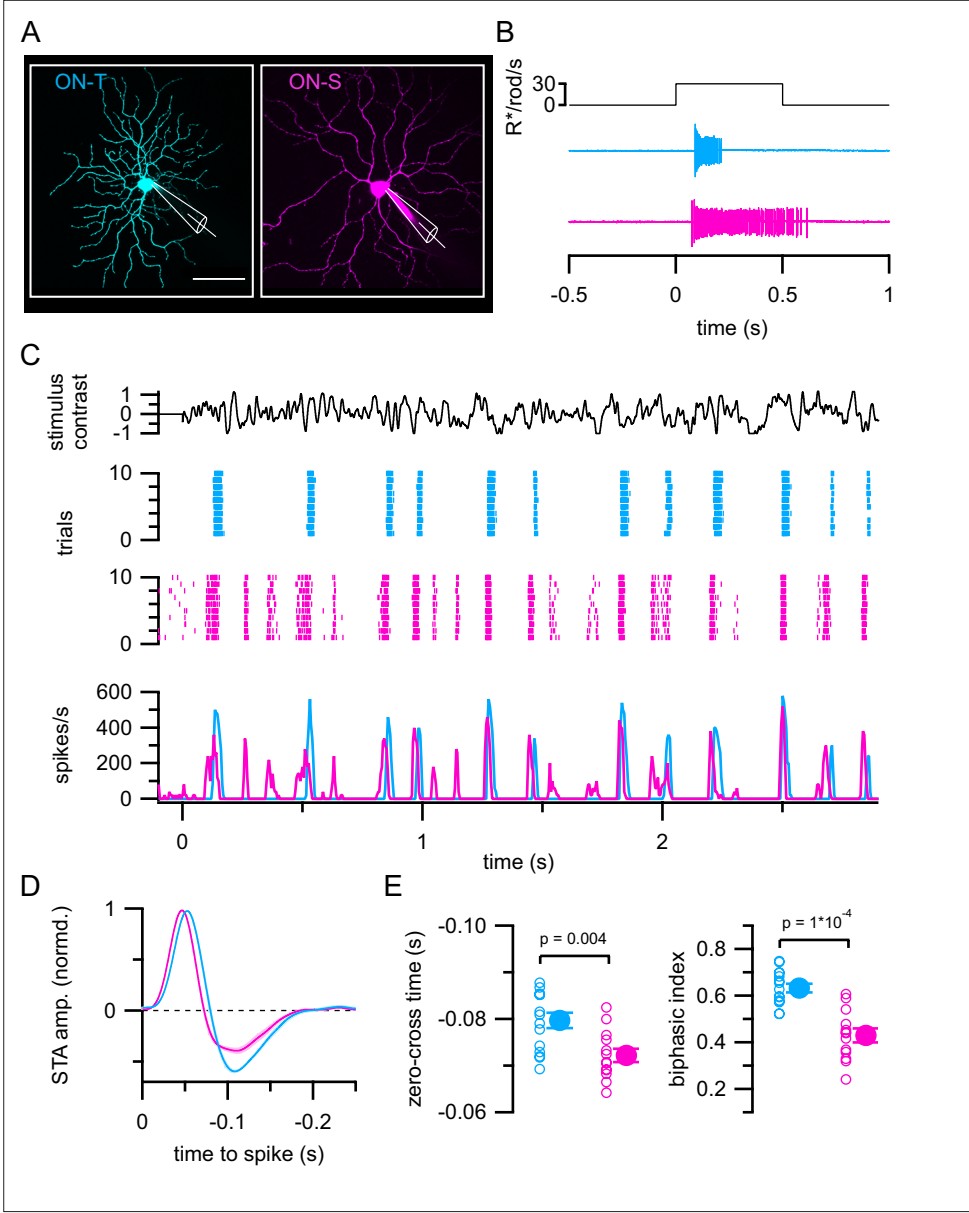

**Figure 1.** Distinct temporal spiking response properties of ON-transient (ON-T) versus ON-sustained (ON-S) retinal ganglion cells (RGCs). (**A**) Maximum intensity projection z-stack fluorescence images of ON-T (left, cyan) and ON-S (right, magenta) RGCs dialyzed with Alexa Fluor 594 during whole-cell patch-clamp recording. Scale bar 50 μm. (**B**) Characteristic spiking responses to a light increment for example ON-T and ON-S RGCs. (**C**) Example spiking responses to repeated trials of spatially uniform (520 μm spot diameter) Gaussian noise stimulus (top black trace) presented at a mean luminance of ~2000 P*/S-cone/s (~3300 R*/rod/s and ~1650 P*/M-cone/s). ON-T (cyan) and ON-S (magenta) RGCs were recorded sequentially in the same piece of retina. Average spike responses (bottom) calculated from 20 repetitions of the noise stimulus. (**D**) ON-T and ON-S spike-triggered averages (STAs) derived from responses to non-repeated noise stimuli. Colored lines and shaded areas show mean ± SEM STA waveforms from ON-T ($n$ = 15) and ON-S ($n$ = 13) RGCs. The STA time axis is reversed to facilitate comparison with linear filters in *Figures 2 and 4*. The mean line obscures the SEM shading for much of the filter in both cell types. (**E**) Quantification of STA kinetics. Open circles show measurements from individual cells and filled circles with error bars show mean ± SEM values. Significance values from Wilcoxon rank sum tests.

The online version of this article includes the following figure supplement(s) for figure 1:

**Figure supplement 1.** Noise-evoked responses are biased toward large amplitude events in ON-T compared to ON-S retinal ganglion cells (RGCs).

ON-transient RGC of *Goetz et al., 2022*. ON-S RGCs correspond to the PV1 RGC type of *Farrow et al., 2013* and 8w RGC type of *Bae et al., 2018*.

We characterized the temporal spiking response properties of ON-T and ON-S RGCs by using an extracellular cell-attached patch electrode to measure action potential firing from either cell type while presenting a spatially uniform, randomly fluctuating light stimulus (Gaussian noise) that allowed us to efficiently probe a variety of stimulus contrasts and temporal frequencies. These and all subsequent physiological recordings were conducted using a photopic mean background light level (~3300 R*/rod/s; ~2000 P*/S-cone/s; and ~1650 P*/M-cone/s) to suppress rod photoreceptor responses and focus on cone-mediated signaling (*Grimes et al., 2018*; *Ke et al., 2014*). *Figure 1C* shows the spiking response of example ON-T and ON-S RGCs to multiple repetitions of the same random stimulus contrast sequence; these RGC types clearly responded differently to temporally modulated light input. In particular, noise-evoked ON-T RGC spike responses were comprised primarily of large amplitude, less frequent spike bursts, whereas ON-S RGC responses were less discrete and consisted of events with a wider range of amplitudes (*Figure 1C*, *Figure 1—figure supplement 1A*).

We summarized the response of each ON-T and ON-S RGC by computing the average stimulus preceding each recorded spike (the spike-triggered average or STA stimulus) when presenting a non-repeated Gaussian noise stimulus (*Figure 1D*; *Chichilnisky, 2001*). We quantified response kinetics from these STA waveforms by measuring the time of zero-crossing (between trough and peak) as well as the ratio of amplitudes of the trough and peak (biphasic index). ON-T RGC STAs were significantly more biphasic (biphasic index = 0.63 ± 0.07; mean ± S.D.; *n* = 15) than ON-S RGC STAs (biphasic index = 0.43 ± 0.11; mean ± S.D.; *n* = 13) (*Figure 1E, right*), consistent with their more transient responses to step stimuli (*Figure 1B*). ON-T RGC responses were also slightly delayed (zero-cross time = –0.080 ± 0.0063 s; mean ± S.D.; *n* = 15) relative to those of ON-S RGCs (zero-cross time = –0.072 ± 0.0052 s; mean ± S.D.; *n* = 13) (*Figure 1C, bottom* and *Figure 1E, left*). The STA provides a convenient summary of RGC temporal response properties, but it does not capture all differences between the RGC spike responses – e.g. the discreteness of the ON-T responses in *Figure 1C*. Nonetheless, the combination of direct observation of the spike responses and STA kinetics illustrates how ON-S and ON-T responses differ.

## Bipolar cell to RGC synapses are an important site of signal diversification in ON pathways

A variety of mechanisms could contribute to the distinct temporal response characteristics of ON-T versus ON-S RGCs, including differences in excitatory and/or inhibitory input conveyed by presynaptic circuits. Parallel visual channels are initially established by the divergence of cone signals to different cone bipolar cell subtypes, which have axons that terminate at distinct depths within the IPL and can correspondingly contact distinct RGC types (*Euler et al., 2014*; *Wässle et al., 2009*). Prior studies have demonstrated that the kinetics of light-evoked glutamate release differ across IPL layers and hence across different bipolar cell synaptic terminals (*Borghuis et al., 2013*; *Franke et al., 2017*). We therefore sought to understand to what extent kinetic differences in bipolar cell-mediated excitatory synaptic input could account for the transient versus sustained responses of these RGCs.

To do this, we used somatic whole-cell voltage-clamp recordings to measure excitatory synaptic currents from both RGC types while stimulating the retina with the same spatially uniform Gaussian noise stimulus that we used to characterize RGC spike output (*Figure 2A*). We used these measurements to construct linear–nonlinear (LN) models comprised of a linear temporal filter (*Figure 2—figure supplement 1A*) and a time-invariant (static) nonlinearity (*Figure 2—figure supplement 1B*) that together provide a simple estimate of the transformation between visual input and excitatory synaptic input to each RGC type (*Kim and Rieke, 2001*). LN models for each RGC type accurately predicted measured responses to noise stimuli that were not used to construct the models, although LN models of ON-S RGC responses performed better than those for ON-T RGCs (correlation coefficient between average response to repeated noise stimulus and model prediction: 0.61 ± 0.06 (ON-S, *n* = 9) and 0.50 ± 0.13 (ON-T, *n*=11), mean ± S.D.; p = 0.028 Wilcoxon rank sum test). ON-T stimulus-evoked excitatory currents were biased toward large amplitude events compared to those of ON-S RGCs (*Figure 1—figure supplement 1B*), like the distinct amplitude distributions of spike events in these cell types noted above (*Figure 1C*, *Figure 1—figure supplement 1A*).

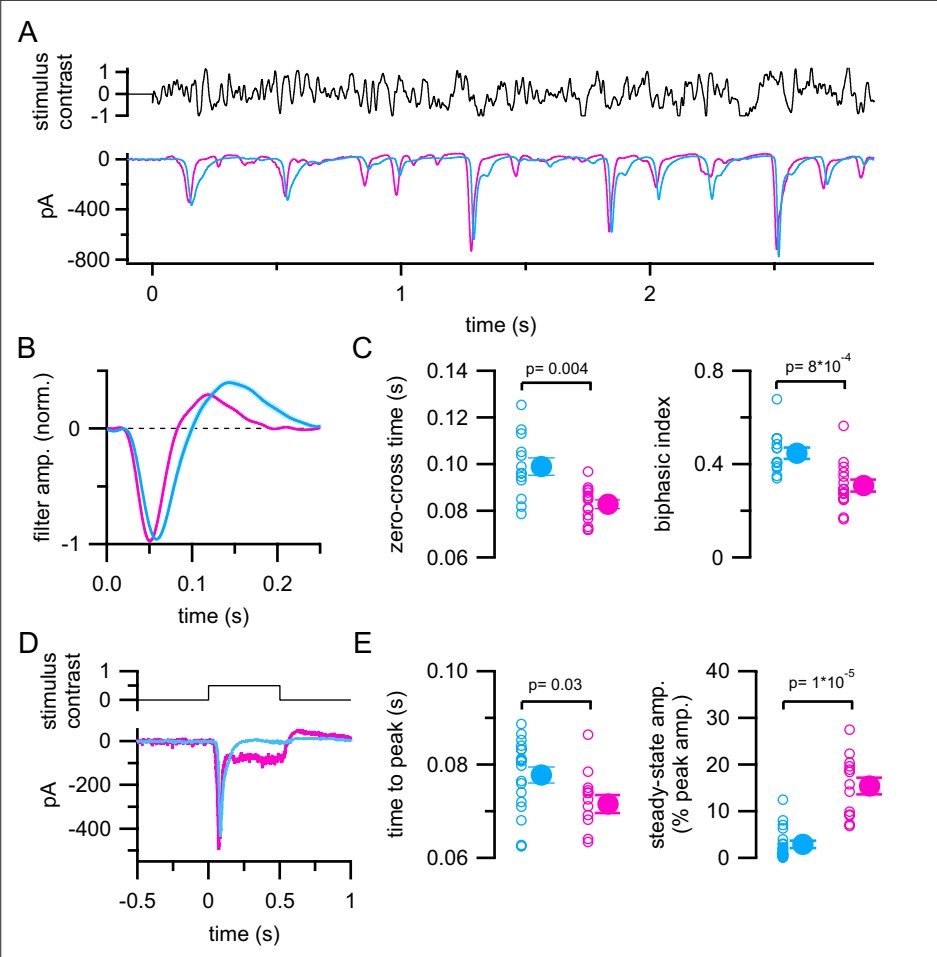

**Figure 2.** Distinct excitatory synaptic input to ON-T and ON-S retinal ganglion cells (RGCs). (**A**) Excitatory synaptic currents measured in example ON-T (cyan) and ON-S (pink) RGCs in response to the same Gaussian noise stimulus used in *Figure 1*. Traces show average currents from 10 repetitions of the same stimulus and are baseline subtracted to facilitate comparison. (**B**) Linear filters computed from responses to non-repeated noise stimulus presentation. Lines and shaded areas show mean ± SEM of responses from ON-T (*n* = 13) and ON-S (*n* = 15) RGCs. The mean line obscures the SEM shading for much of the filter in both cell types. (**C**) Quantification of filter kinetics. (**D**) Excitatory currents measured from example ON-T and ON-S RGCs in response to a 0.5-s step increase in light intensity (50% contrast; mean response to five repetitions of stimulus). (**E**) Quantification of step response kinetics (*n* = 19 ON-T RGCs; *n* = 13 ON-S RGCs). In (**C**) and (**E**), open circles show measurements from individual cells and filled circles with error bars show mean ± SEM values. Significance values from Wilcoxon rank sum tests.

The online version of this article includes the following figure supplement(s) for figure 2:

**Figure supplement 1.** Linear–nonlinear models of excitatory synaptic input to ON-T and ON-S retinal ganglion cells (RGCs).

**Figure supplement 2.** Inhibitory synaptic input to ON-T and ON-S retinal ganglion cells (RGCs).

The linear filter captures the kinetics of the RGC response and is analogous to the time-reversed STA derived from spike responses (*Chichilnisky, 2001*). As with the STAs in *Figure 1D*, excitatory linear filters were slower and more biphasic in ON-T (zero-cross time = 0.10 ± 0.013 s; biphasic ind. = 0.45 ± 0.090; mean ± S.D.; *n* = 13) compared to ON-S RGCs (zero-cross time = 0.083 ± 0.0074 s; biphasic ind. = 0.31 ± 0.10; mean ± S.D.; *n* = 15) (*Figure 2C*). This latter difference is consistent with more transient light-evoked excitatory synaptic input to ON-T RGCs. Indeed, light step-evoked excitatory currents to ON-T RGCs decayed to baseline levels within a few hundred milliseconds, even in the continued presence of light input, whereas excitatory inputs to ON-S RGCs had a sustained component that persisted for the duration of the light step (*Figure 2D, E*).

These results show that transient versus sustained kinetics are already present in the excitatory pathways that provide input to ON-T and ON-S RGCs. However, these findings do not rule out additional contributions to the distinct spiking behavior of these RGC types from amacrine cell-mediated inhibitory synaptic input to the RGCs or from different intrinsic spiking properties. To examine whether postsynaptic inhibition contributes importantly to kinetic differences between ON-T and ON-S RGCs, we measured both excitatory and inhibitory synaptic currents in response to Gaussian noise stimuli in a subset of RGC recordings (*Figure 2—figure supplement 2*). Like excitatory currents, light-evoked inhibitory currents were more transient in ON-T RGCs (linear filter biphasic ind. = 0.34 ± 0.12; mean ± S.D.; *n* = 8) compared to those in ON-S RGCs (biphasic ind. = 0.16 ± 0.09; mean ± S.D.; *n* = 8) (*Figure 2—figure supplement 2B, C*; p = 0.01, Wilcoxon rank sum test). We also quantified the relative strength of excitatory and inhibitory synaptic inputs by calculating the ratio between the largest noise-evoked excitatory and inhibitory conductances ($G_{exc}$ and $G_{inh}$; derived from the measured currents) in response to a repeated noise stimulus in each RGC (*Figure 2—figure supplement 2A, D, E*). This ratio did not differ significantly between the two cell types (ON-T RGCs: peak $G_{exc}$ /peak $G_{inh}$ = 1.46 ± 0.65; mean ± S.D.; *n* = 6; ON-S RGCs: peak $G_{exc}$ /peak $G_{inh}$ = 4.04 ± 2.48; mean ± S.D.; *n* = 7; p = 0.10, Wilcoxon rank sum test). The relatively small inhibitory inputs, together with the clear differences we observed in excitatory synaptic input between ON-T and ON-S RGCs, indicate that bipolar cell pathways have a central role in establishing transient versus sustained kinetics.

## ON-T and ON-S RGCs receive input from distinct bipolar types

Retinal circuits dedicated to transient versus sustained signaling are generally located at the center and borders of the IPL, respectively (*Baden et al., 2016*; *Bae et al., 2018*; *Beaudoin et al., 2019*; *Borghuis et al., 2013*; *Franke et al., 2017*; *Roska and Werblin, 2001*). We compared stratification patterns of ON-T and ON-S RGCs in a subset of whole-cell recordings by filling cells with a fluorescent dye and imaging the dendrites of physiologically characterized RGCs using two-photon microscopy. We assessed IPL depth in live tissue by using retinas from a transgenic mouse line in which type 5 bipolar cells strongly express EGFP (*Gjd2-EGFP* mice). These EGFP+ bipolar cells express calcium binding protein 5 (CaBP5; *Figure 3—figure supplement 1B–D*) and have axons that occupy a narrow band just above and overlapping with the dendrites of ON starburst amacrine cells (*Figure 3—figure supplement 1A*; *Ghosh et al., 2004*). Based on the regular spacing of their somas, these cells likely correspond to one of the three subtypes of type 5 bipolar cells previously identified in mouse retina (*Greene et al., 2016*; *Rodieck, 1991*; *Figure 3—figure supplement 1E, F*). Imaging of RGCs together with EGFP+ axon terminals in *Gjd2-EGFP* mice revealed that ON-T and ON-S RGC dendrites occupy distinct, mostly non-overlapping regions of the IPL (*Figure 3—figure supplement 2*).

The distinct IPL stratification of ON-T and ON-S RGCs indicates they receive input from different bipolar cell populations. Recent serial block face scanning electron microscopy (SBFSEM) connectomic analyses found that type 6 and 7 bipolar cells account for ~50–52% and 46% of the excitatory synapses formed upon ON-S RGC dendrites (*Sabbah et al., 2018*; *Swygart et al., 2024*). However, the bipolar cell subtype(s) presynaptic to ON-T RGCs have not previously been identified. Our observation that ON-T RGC dendrites occupy a narrow band in the IPL that closely matches the stratification pattern of EGFP+ type 5 bipolar cell axons in *Gjd2-EGFP* mice (*Figure 3—figure supplement 2*) suggests that these bipolar cells likely provide a substantial fraction of the excitatory input to ON-T RGCs. To test this possibility, we transfected retinas from *Gjd2-EGFP* mice with a plasmid encoding PSD-95 fused to the red fluorescent reporter mCherry to label postsynaptic sites of excitatory input (*Figure 3—figure supplement 3*). We used confocal microscopy to quantify the fraction of putative synapses, defined as sites with overlapping PSD-95 and EGFP fluorescent signals, that arose from *Gjd2-EGFP+* axon terminals (*Schwartz et al., 2012*). From this analysis, we concluded that the labeled type 5 bipolar cells in *Gjd2-EGFP* mice provide ~40% of the ribbon synaptic input to ON-T RGCs (*Figure 3—figure supplement 3D*).

We next performed correlative fluorescence imaging and SBFSEM to identify which type 5 bipolar subtype is labeled in *Gjd2-EGFP* mice and to further characterize what other bipolar cell types provide the remaining input to ON-T RGCs. We first used 2P microscopy to image the dendritic morphology of two dye-filled ON-T RGCs that were identified from their light-evoked spike responses in flat-mounted retina from *Gjd2-EGFP* transgenic mice (*Figure 3A*). After 2P imaging, we fixed the retina and used the near-infrared branding (NIRB) technique to create fiducial marks in the tissue around

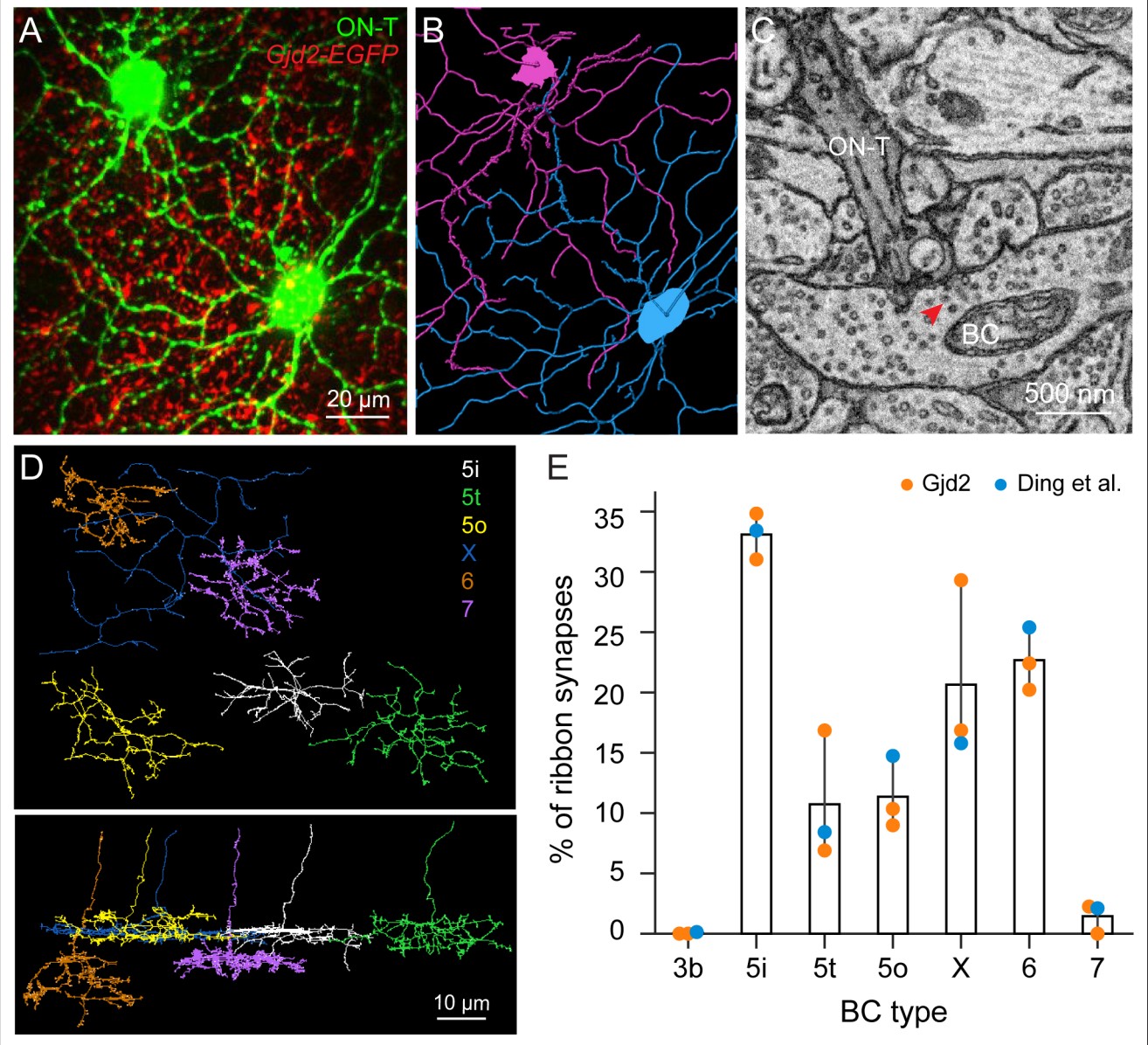

**Figure 3.** Identification of bipolar cell subtypes presynaptic to ON-T retinal ganglion cells (RGCs). (**A**) Maximum-projection z-stacks of two dye-filled ON-T RGCs (green) in whole-mount retina from *Gjd2-EGFP* (red) transgenic mice. (**B**) Traced skeletons of dendritic arbors of the two ON-T RGCs from EM volume. (**C**) An EM micrograph showing a ribbon synapse (arrowhead) on a BC axonal terminal onto the dendrite of a ON-T RGC. (**D**) An example skeleton for each presynaptic BC type (types 5i, 5t, 5o, X, 6, and 7) of ON-T RGCs. Top: whole-mount view, bottom: side view. (**E**) Bar plot showing the proportion of input synapses to ON-T RGCs from each BC type. Mean ± SEM (*n* = 3 RGCs, 2 from this study and 1 from ***Ding et al., 2016***).

The online version of this article includes the following figure supplement(s) for figure 3:

**Figure supplement 1.** Characterization of *Gjd2-EGFP+* bipolar cells.

**Figure supplement 2.** ON-T and ON-S retinal ganglion cell (RGC) dendrites ramify in distinct inner plexiform layer (IPL) sublaminae.

**Figure supplement 3.** *Gjd2-EGFP+* type 5 bipolar cells provide ~40% of total synapses onto ON-T retinal ganglion cells (RGCs).

**Figure supplement 4.** Classification of three subtypes of type 5 bipolar cells.

the dye-filled ON-T RGCs (***Bishop et al., 2011***). After acquiring the SBFSEM volume, we located the somas of the two ON-T RGCs using these fiducial marks and reconstructed the dendritic arbors of both cells (e.g. ***Figure 3B***). The dendritic morphologies from our EM reconstruction matched those from the light microscopy images. This suggests that we successfully identified the two ON-T RGCs recorded physiologically in our EM volume. We chose the two longest dendritic arbors from the RGC

soma to the dendritic tip and identified all the synapses from bipolar cells along these dendrites (*Figure 3C*). We then reconstructed all the presynaptic bipolar cells and classified them according to their axon morphology and stratification (*Figure 3D*). Careful examination of axonal tiling among type 5i, 5o, and 5t subtypes provided additional confirmation of the classification of these type 5 bipolar cells (*Figure 3—figure supplement 4A*; *Greene et al., 2016*). When comparing the EM images with light microscopy images, it is clear that *Gjd2-EGFP* bipolar cells are type 5i bipolar cells (*Figure 3—figure supplement 4B*).

Quantification of bipolar inputs to ON-T RGCs based on the SBFSEM reconstructions reinforced results from the light microscopy approach described above. Type 5i bipolar cell axon terminals provided the largest fraction of synapses onto the two ON-T RGCs within our SBFSEM volume (~30–35%). Other inputs arose from type 5o (~10%), type 5t (~5–15%), XBC (~15–30%), and, somewhat surprisingly given the differences in stratification, from type 6 (~20%) bipolar cells (*Figure 3E*). We confirmed this connectivity pattern by additionally analyzing bipolar cell synaptic inputs to a putative ON-T RGC (RGC7089) in the publicly available SBFSEM volume (*Ding et al., 2016*) that was used by *Sabbah et al., 2018* to map inputs to ON-S RGCs (*Figure 3E*).

## Bipolar cell light responses differ minimally across types

Are transient versus sustained signals already differentiated in the voltage responses of the different bipolar cells presynaptic to ON-T and ON-S RGCs? Or is the distinct excitatory synaptic input that these RGCs receive primarily due to differences at the bipolar cell to RGC synapse? To answer these questions, we compared the light-driven response properties of the bipolar cell subtypes that comprise most of the excitatory input to ON-T or ON-S RGCs. We made targeted electrophysiological recordings from type 5i, 6, and 7 bipolar cells in retinal slices from three transgenic mouse lines in which these different bipolar cell subtypes express fluorescent proteins (see Methods). In these experiments, we used somatic whole-cell perforated-patch recordings to limit perturbation of the G-protein signaling cascade that mediates ON bipolar cell light responses.

Somatic resting membrane potentials did not differ significantly among these bipolar cell types (mean ± S.D. $V_{rest}$: −38 ± 6.3 mV in type 5i ($n$ = 11); −33.7 ± 6.3 mV in type 6 ($n$ = 7); −36.5 ± 4.7 mV in type 7 ($n$ = 6); Kruskal–Wallis test, p = 0.35; values are not corrected for the liquid junction potential since that is hard to estimate in perforated-patch recordings). We characterized light responses using the same spatially uniform Gaussian noise stimulus (*Figure 4B*) and LN model analysis (*Figure 4C*, *Figure 4—figure supplement 1*) as in RGC recordings to facilitate direct comparison of results from bipolar cells and RGCs (*Kim and Rieke, 2001*; *Rieke, 2001*). LN models of bipolar cell responses accurately predicted responses to noise stimuli (correlation coefficient between average response to repeated noise stimulus and LN model prediction = 0.75 ± 0.15 for $n$ = 18 bipolar cells ($n$ = 13 type 5; $n$ = 3 type 6; $n$ = 2 type 7)). Unlike for ON-T and ON-S RGCs, temporal characteristics of type 5i, 6, and 7 bipolar cell linear filters were indistinguishable from each other under our recording conditions (*Figure 4B–D*). Although there was some variability in the temporal response properties across the population of bipolar cells we recorded, these differences were not associated with morphologically and molecularly defined bipolar cell types. Rather, we observed variability in response kinetics within each bipolar cell subtype, and temporal responses thus overlapped substantially across different type 5i, 6, and 7 bipolar cells (*Figure 4*, *Figure 4—figure supplement 1*).

Responses to light steps measured in a subset of type 5i and 6 bipolar cell recordings confirmed our conclusion from Gaussian noise stimuli that different bipolar cell types exhibit nearly identical light response kinetics under our recording conditions (*Figure 4E, F*). Unlike the clear differences in light step-evoked excitatory currents we measured in ON-T and ON-S RGCs (*Figure 2D, E*), type 5i and 6 bipolar cells both had sustained responses to light steps. These findings indicate that mechanisms upstream of the generation of the cone bipolar cell voltage response, such as synaptic transmission from cones to cone bipolar cells and intrinsic response properties of the cone bipolar cells, cannot explain the kinetically distinct responses in parallel ON channels mediated by type 5i, 6, or 7 bipolar cells. Consistent with this conclusion, ON-T and ON-S RGCs exhibit kinetically distinct light step-evoked spiking responses under scotopic conditions (not shown), where much of the RGC response originates from signals carried from outer to inner retina via the rod bipolar cell pathway (*Figure 1B*; *Grimes et al., 2014*; *Ke et al., 2014*). Because AII amacrine cells distribute rod bipolar cell signals

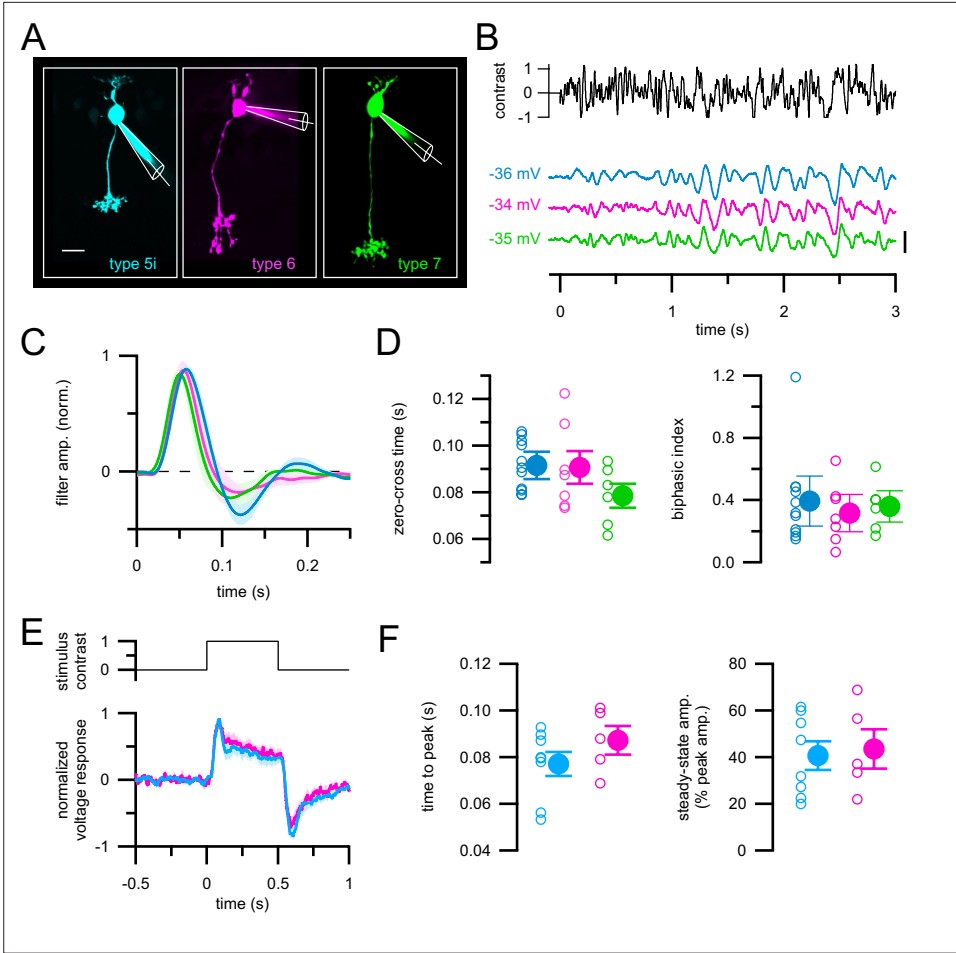

**Figure 4.** Bipolar cell subtypes presynaptic to ON-T and ON-S retinal ganglion cells RGCs have indistinguishable temporal response properties. (**A**) Maximum-projection z-stacks of fluorescent dye-filled bipolar cells targeted for electrophysiological recording in retinal slices from *Gjd2-EGFP* (left), *Grm6-tdTomato* (middle), and *Gus8.4-EGFP* (right) transgenic mice. (**B**) Responses to spatially uniform Gaussian noise stimulation (same as in ***Figures 1 and 2***) in example type 5i (cyan), type 6 (magenta), and type 7 (green) bipolar cells. Voltage traces are average responses to 10 repetitions of the noise stimulus. Scale bar = 5 mV. (**C**) Linear filters computed from responses to non-repeated noise stimulus presentation. Lines and shaded areas show mean ± SEM of responses (type 5i, *n* = 11; type 6, *n* = 7; type 7, *n* = 6). (**D**) Quantification of response kinetics from linear filters. No significant differences across types (p = 0.19 zero-cross times; p = 0.89 biphasic index; Kruskal–Wallis test). (**E**) Responses to step light increment. Lines and shaded areas show peak-normalized mean ± SEM responses across type 5i bipolar cells (*n* = 8) and type 6 bipolar cells (*n* = 5). (**F**) Quantification of step response kinetics. Time to peak and steady-state response amplitude were not different between type 5i and 6 bipolar cells (p = 0.72 and p = 0.35, respectively; Wilcoxon rank sum test).

The online version of this article includes the following figure supplement(s) for figure 4:

**Figure supplement 1.** Linear–nonlinear models of bipolar cell voltage responses.

directly to cone bipolar cell axon terminals, differences observed under scotopic conditions likely arise at and/or downstream of cone bipolar cell synaptic output.

## Presynaptic inhibition does not establish kinetic differences in bipolar cell-mediated excitation

Bipolar cells participate in diverse inhibitory circuits with amacrine cells, which are comprised of >40 morphologically and functionally distinct cell types (***Diamond, 2017***; ***Helmstaedter et al., 2013***). Past work suggests that amacrine cell-mediated presynaptic inhibition is critical to bipolar cell subtype-specific patterns of light-evoked glutamate release (***Franke et al., 2017***). Our bipolar cell voltage

recordings, which should reflect a combination of light-evoked excitatory and inhibitory input, argue that presynaptic inhibition is not required for generating differential synaptic output between type 5i, 6, and 7 bipolar cells. However, the slicing procedure we used to prepare tissue for bipolar cell recordings may damage some inhibitory connections, especially long-range amacrine cell inputs. Distance-dependent attenuation of voltage signals between bipolar cell axon terminals and the somatic recording site could also limit our ability to accurately measure the influence of presynaptic inhibition in these recordings. We therefore performed additional anatomical and pharmacological experiments to investigate a possible role for presynaptic inhibition in generating kinetically distinct excitation to ON-T and ON-S RGCs.

Many bipolar cell output synapses are organized into a 'dyad' arrangement in which a bipolar cell ribbon synapse releases glutamate onto the processes of two different postsynaptic cells. At dyads, bipolar cell output synapses can be presynaptic to the processes of two amacrine cells, two RGCs, or an amacrine cell and a RGC (*Cohen and Sterling, 1990*; *Dowling and Boycott, 1966*; *McGuire et al., 1984*; *Yu et al., 2023*). Dyad synapses comprised of an amacrine cell and a RGC may shape bipolar cell glutamate release dynamics onto the RGC partner because glutamatergic excitation of the postsynaptic amacrine cell can elicit release of inhibitory neurotransmitters (GABA or glycine) from the amacrine cell processes that feed back onto the bipolar cell axon terminal. This form of inhibitory feedback control of bipolar cell output, termed reciprocal inhibition, could potentially contribute to differences in glutamate release onto ON-T and ON-S RGCs.

We analyzed bipolar cell dyads with postsynaptic amacrine cell processes and either ON-T or ON-S RGC dendrites in SBFSEM volumes to examine whether reciprocal inhibition is likely to generate more transient glutamate release onto ON-T RGCs. We defined putative reciprocal synapses as bipolar cell/amacrine cell/RGC dyads in which we observed clustered vesicles within the amacrine cell process that were in close proximity to the bipolar cell ribbon synapse (*Figure 5A*). Our ultrastructural analysis revealed that the prevalence of reciprocal synapses was similar at dyad synapses with postsynaptic ON-T versus ON-S RGCs (*Figure 5A, B*). This finding is not consistent with an important role for feedback/reciprocal inhibition in generating differential bipolar cell output onto the different RGC types.

We also performed pharmacological experiments to further examine what role, if any, inhibition may have in establishing transient glutamate release onto ON-T RGC dendrites. We used artificial cerebrospinal fluid (ACSF) in place of Ames solution for the dissection and maintenance of tissue for these experiments. ACSF is often used in retina experiments, including the glutamate release measurements described below, and we wanted to check that our results generalized across conditions. Light step-evoked currents in both ON-T and ON-S RGCs were more sustained than responses measured using Ames' solution (*Figure 5—figure supplement 1A, B*). Nonetheless, the difference between sustained and transient RGC types was maintained across different experimental conditions.

In ON-T RGCs, blockade of GABA$_A$ and GABA$_C$ receptors via bath co-application of SR95531 and TPMPA enhanced the peak amplitude of the initial transient component of light step-evoked excitatory synaptic currents (–2023 ± 372 pA; mean ± S.D.) compared to control (–834 ± 103 pA) (p = 0.031, Wilcoxon sign rank test; $n$ = 6) whereas the sustained component at the end of the light step was not significantly affected (control: –201 ± 15 pA; GABA-R block: –291 ± 44; p = 0.16, Wilcoxon sign rank test; $n$ = 6) (*Figure 5C*). The strong enhancement of the initial phase of the response we observed is consistent with prior work showing that GABA$_C$ receptor-mediated inhibition limits multivesicular release from bipolar cells (*Sagdullaev et al., 2006*). Block of GABA receptors thus made excitatory currents in ON-T RGCs more transient compared to control conditions (*Figure 5D*). Note that this is opposite the expected effect if presynaptic GABA receptors contribute to generating transient responses in these cells. GABA receptor antagonists should have made excitatory currents more sustained if presynaptic GABAergic inhibition was generating transient bipolar cell input to ON-T RGCs. Thus, GABAergic inhibition appears to control response gain and time course of bipolar cell synaptic output to ON-T RGCs but is not responsible for the transient kinetics of excitatory input to these cells.

We also found that glycinergic inhibition is not likely to contribute importantly to the more transient glutamate release from bipolar cells presynaptic to ON-T RGCs; bath application of the glycine receptor antagonist strychnine did not significantly alter the kinetics of light step-evoked excitatory currents in these cells (*Figure 5E, F*). Taken together with our ultrastructural analysis, these results

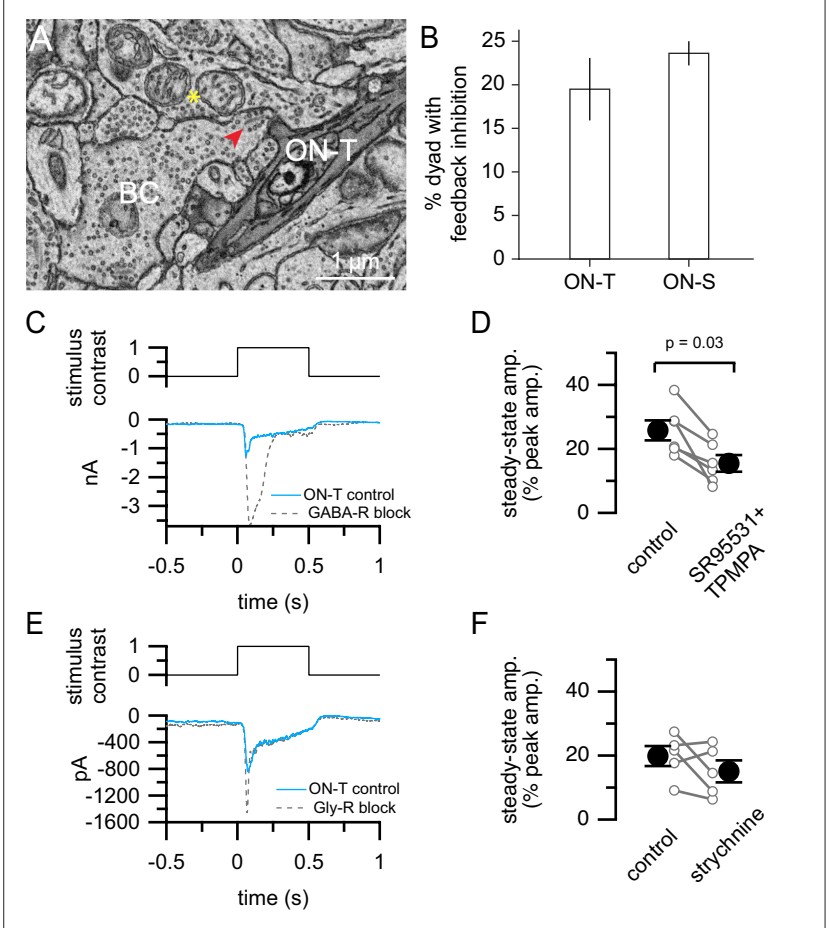

**Figure 5.** Presynaptic inhibition is not required for kinetic differences between bipolar cell inputs to ON-T versus ON-S retinal ganglion cells (RGCs). (**A**) Example EM micrograph showing a ribbon dyad (red arrowhead) between a BC and an ON-T RGC and an amacrine cell, with an adjacent feedback inhibitory synapse (yellow asterisk). (**B**) Bar plots showing the percentage of RGC/AC ribbon dyads with feedback inhibition. Mean ± S.D. ($n$ = 2 ON-S RGCs 85 dyads analyzed, 2 ON-T RGCs with 83 dyads analyzed). (**C**) Excitatory currents measured in response to light step increment in control conditions (cyan) and in the presence of GABA receptor antagonists (10 µM SR95531 and 50 µM TPMPA; dashed gray) in an example ON-T RGC. Traces are mean response from six repeats of stimulus presentation. (**D**) Summary of steady-state amplitudes measured from step responses in control or with GABA receptor antagonists. Step-evoked currents were significantly more transient when GABA$_{A/C}$ receptors were blocked (p = 0.031; Wilcoxon signed rank test; $n$ = 6 RGCs). (**E**) Control excitatory currents (cyan) or with addition of strychnine (1 µM; dashed gray) to bath in an example ON-T RGC. (**F**) Summary of strychnine experiments. No significant difference between conditions (p = 0.44; Wilcoxon signed rank test; $n$ = 5 RGCs). In (**D**) and (**F**), open circles with lines show measurements from individual ON-T RGCs. Filled circles with error bars show mean ± SEM.

The online version of this article includes the following figure supplement(s) for figure 5:

**Figure supplement 1.** Presynaptic inhibition does not affect response kinetics but does affect spatial integration for excitatory inputs to ON-T retinal ganglion cells (RGCs).

show that amacrine cell-mediated GABAergic or glycinergic inhibition of bipolar cell axon terminals does not explain kinetic differences in excitatory synaptic input to ON-T and ON-S RGCs.

## Bipolar cell glutamate release differs across types

Taken together, the bipolar cell and RGC recordings in *Figure 2* and *Figure 4* indicate that bipolar cell synapses have a key role in the separation of visual signals into transient or sustained channels within ON pathways in the retina. Presynaptic and/or postsynaptic mechanisms could contribute to the distinct kinetics of bipolar cell-mediated excitatory input to ON-T and ON-S RGCs. To investigate whether presynaptic differences in glutamate release from bipolar cell axon terminals shape

RGC response kinetics, we expressed the fluorescent glutamate reporter iGluSnFR on the dendrites of ON-T and ON-S RGCs (see Methods). Using two-photon glutamate sensor imaging, we found that light-evoked glutamate signals had significantly more transient kinetics at ON-T RGC dendrites (steady-state amplitude as % of peak amplitude = 21 ± 13; mean ± S.D.; $n$ = 224 ROIs) compared to ON-S RGC dendrites (steady-state amp. as % of peak amp. = 44 ± 12; mean ± S.D.; $n$ = 307 ROIs) (*Figure 6*).

The experiments summarized in *Figure 5* suggest that presynaptic inhibitory input to bipolar axon terminals is not required for the difference between ON-T and ON-S responses. The glutamate release measurements provided a further test. We compared the kinetics of release before and after suppressing spiking amacrine cell activity (with bath application of the voltage-gated $Na^+$ channel blocker TTX) or all amacrine cells (with bath application of the glutamate receptor antagonist NBQX). This approach avoided the confounding changes in excitatory signals we encountered with inhibitory receptor blockers (e.g. the increase in excitatory input to ON-T RGCs in *Figure 5C* and oscillatory bursts of excitation in ON-S RGCs (not shown)). The more transient glutamate release kinetics at ON-T versus ON-S dendrites remained even when amacrine cell responses were suppressed (control: steady-state amp. as % of peak amp. 13 ± 10; mean ± S.D.; $n$ = 189 ROIs/4 FOVs for ON-T dendrites vs. 40 ± 12; mean ± S.D.; $n$ = 287 ROIs/8 FOVs for ON-S dendrites; TTX: 16 ± 9; mean ± S.D.; $n$ = 50 ROIs/3 FOVs for ON-T dendrites vs. 35±11; mean ± S.D.; $n$ = 146 ROIs/6 FOVs for ON-S dendrites; *$p$ < 0.001; $t$-test; NBQX: 7 ± 3; mean ± S.D.; $n$ = 112 ROIs/1 FOV for ON-T dendrites vs. 24 ± 9; mean ± S.D.; $n$ = 97 ROIs/2 FOVs for ON-S dendrites; *$p$ < 0.001; $t$-test; *Figure 7*).

These results show that differences in glutamate release from presynaptic bipolar cells contribute to transient versus sustained ON RGC responses and that the differences in release are retained without amacrine feedback to bipolar cells. These experiments do not, however, rule out the possibility that additional postsynaptic mechanisms, for example, differential glutamate receptor expression or differences in spike generation, contribute to kinetic differences between RGC types.

Although whole-cell excitatory currents in ON-T and ON-S RGCs (*Figure 2*, *Figure 5—figure supplement 1*) and iGluSnFR signals measured from ON-T and ON-S RGC dendrites (*Figure 6*) both exhibited clear kinetic differences between the RGC types, there were quantitative differences in the response to light steps between the different measurement approaches. As described above, these differences appear to arise from differences in the external solution (Ames vs. ACSF). The iGluSnFR experiments used an ACSF solution, like the patch-clamp experiments in *Figure 5*, *Figure 5—figure supplement 1*. Both iGluSnFR responses and excitatory synaptic inputs measured in ACSF showed a larger sustained component of the step response at both ON-T and ON-S dendrites compared to currents measured in Ames (compare *Figure 2D, E*, *Figure 5—figure supplement 1A, B* and *Figure 6B, C*). This observation highlights the importance of comparing responses between different RGC types, as well as between RGCs and bipolar cells, under identical recording conditions.

Importantly, the kinetics of glutamate release measured using iGluSnFR resemble the excitatory synaptic inputs to ON-T and ON-S RGCs more closely than the relevant bipolar voltage responses (*Figure 4*). This indicates that the difference between the ON-T and ON-S responses originates at least in part in the conversion of bipolar voltage to glutamate release.

## Ribbon size and vesicle replenishment

Prior work has shown that bipolar cells can exhibit profound short-term depression of glutamate release due to use-dependent depletion of the readily releasable pool of synaptic vesicles (*Burrone and Lagnado, 2000*; *Singer and Diamond, 2006*; *Singer and Diamond, 2003*; *von Gersdorff and Matthews, 1997*). Strong vesicle depletion at the cone bipolar cell synapses onto ON-T RGCs could generate more transient responses in these cells compared to ON-S RGCs. To examine this possibility, we measured excitatory synaptic inputs to ON-T and ON-S RGCs in response to a paired-flash stimulus in which two brief light flashes were separated by varying time intervals. As in *Figures 1, 2, and 4*, we used Ames' medium for tissue preparation and maintenance in these experiments. These experiments used the same photopic background light level (~3300 R*/rod/s; ~2000 P*/S-cone/s; and ~1650 P*/M-cone/s) as used for Gaussian noise and step stimuli.

Flash-evoked excitatory synaptic currents were suppressed in both RGC types at short inter-flash intervals, consistent with incomplete recovery from vesicle depletion at the time of the second flash (*Figure 8A, top/middle, B*). This suppression was much stronger for currents measured in ON-T RGCs

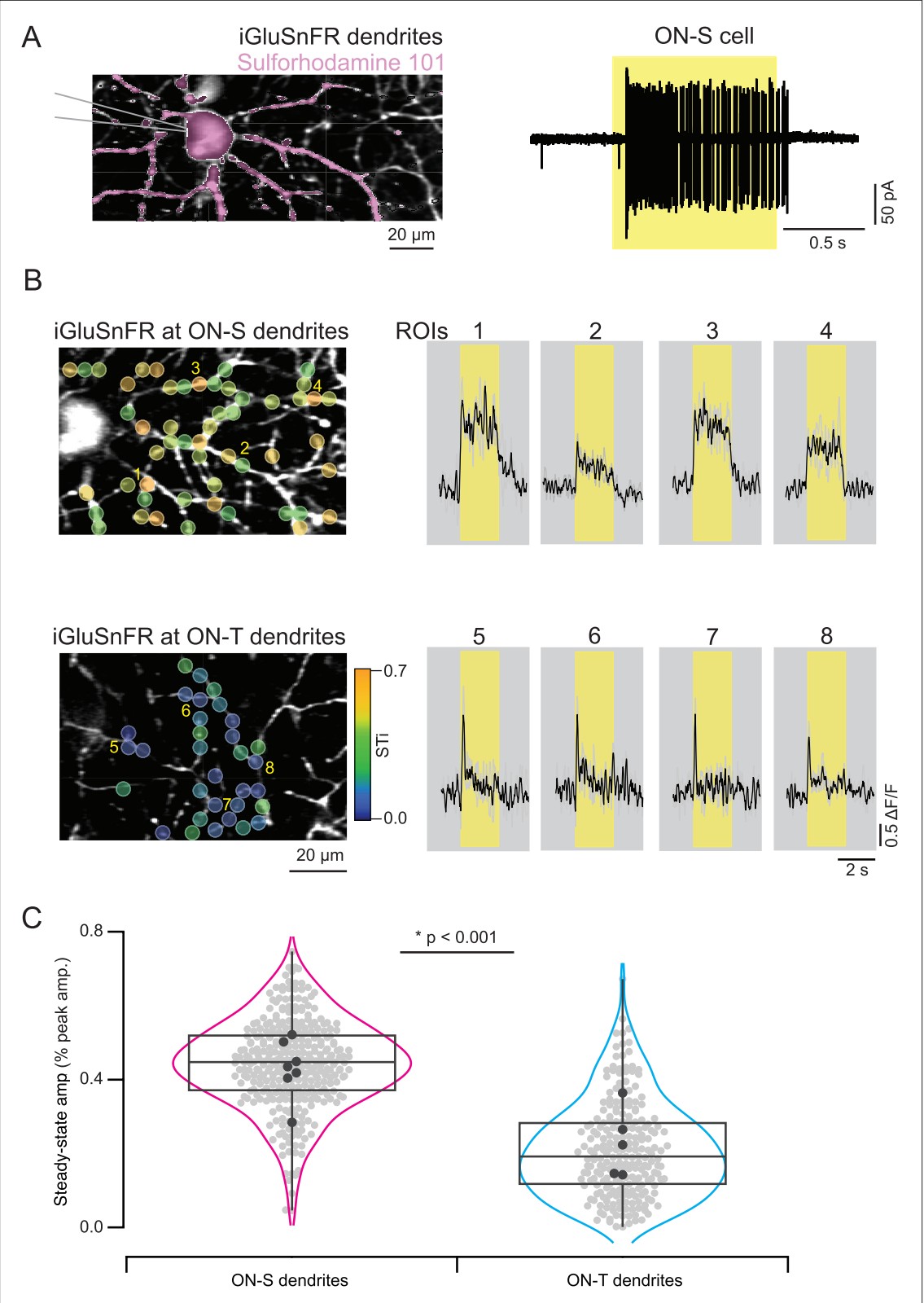

**Figure 6.** Distinct glutamate input kinetics at ON-S versus ON-T retinal ganglion cell (RGC) dendrites. (**A**) ON-S RGC dendrites expressing iGluSnFR in a *Kcng4-Cre* mouse were filled with sulforhodamine 101 (left) after measuring their spiking response to a 200-μm spot extracellularly (right). Yellow bands indicate stimulus duration. (**B**) iGluSnFR expressing ON-S and ON-T dendrites at different depths of the inner-plexiform layer (Left) were imaged using two-photon microscopy. Example iGluSnFR signals extracted from small regions of interest (ROIs 1–8) from ON-S and ON-T RGC dendrites illustrate the

*Figure 6 continued*

kinetic differences in their respective inputs. Black, mean responses; gray, individual trials. (**C**) Distribution of steady-state response amplitudes (mean fluorescence during last 1 s of stimulus) of the individual (gray) and average (black) ROIs from ON-S and ON-T dendrites evoked by a 200-µm static spot (*n* = 7 FOVs, 6 retinas for ON-S and 5 FOVs, 5 retinas for ON-T RGC dendrites, *p < 0.001; *t*-test between ROIs).

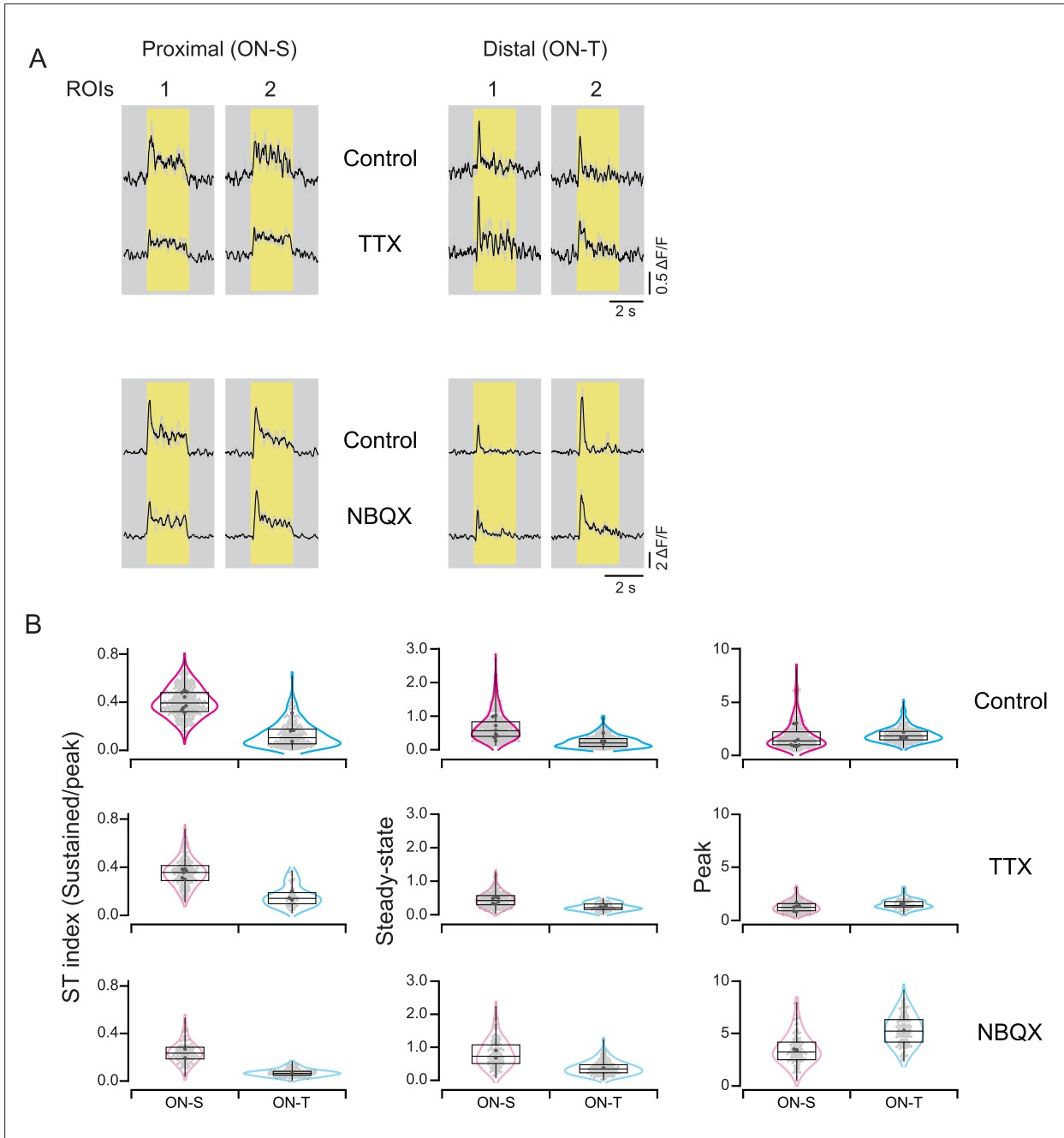

**Figure 7.** The distinction between sustained and transient responses remains when amacrine cell activity is blocked. (**A**) iGluSnFR measurements of glutamate release in control conditions and in the presence of TTX (which blocks spike-dependent amacrine cell output); or in the presence of NBQX, which blocks AMPA receptor-mediated input to all amacrine cells (as well as horizontal cells). (**B**) Ratio of steady state to transient response across ROIs before (top) and during (bottom) application of TTX/NBQX.

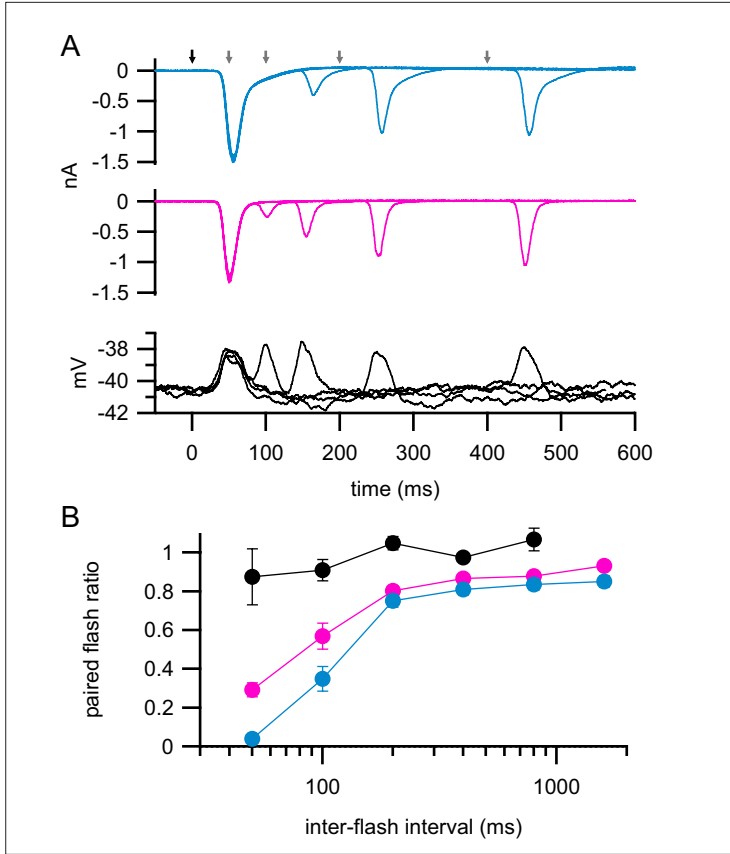

**Figure 8.** Stronger flash-evoked suppression of excitatory currents in ON-T versus ON-S retinal ganglion cells (RGCs). (**A**) Paired flash-evoked excitatory currents measured at different inter-flash intervals (50, 100, 200, and 400 ms) in an example ON-T (top; cyan) and an ON-S (middle, magenta) RGC. Bottom black traces show voltage responses to the same stimulus measured in an example type 5i bipolar cell. Arrows at the top indicate timing of the first flash (black) and second flashes (gray). Flashes were 400% contrast and 10-ms duration. Traces show mean responses to five repeats of each paired-flash interval. (**B**) Filled circles with error bars show mean ± SEM paired-flash ratios at different inter-flash intervals across all ON-T RGCs (cyan; *n* = 6), ON-S RGCs (magenta; *n* = 5), and type 5i bipolar cells (black; *n* = 6).

compared to ON-S RGCs. We quantified the extent of suppression using a paired-flash ratio, which was calculated as the peak amplitude of the response to the second flash divided by the peak amplitude of the response to the first flash. At the shortest interval that we tested (50 ms), the response to a second flash was almost completely suppressed in ON-T RGCs (paired-flash ratio = 0.04 ± 0.03; mean ± S.D.; *n* = 6) whereas the response to a second flash in ON-S RGCs was suppressed considerably less (paired-flash ratio = 0.29 ± 0.08; *n* = 5). To rule out the possibility that a reduction in bipolar cell voltage response to the second flash could account for the strong paired-flash suppression of excitatory input to RGCs, we also measured type 5i bipolar cell responses to the paired-flash stimuli. The amplitude of flash-evoked voltage responses in type 5i bipolar cells was unchanged by a preceding flash at all time intervals (*Figure 8A, bottom, C*), which argues that paired-flash suppression arises at the bipolar output synapse. These results are consistent with the idea that differences in the extent of stimulus-evoked vesicle pool depletion in bipolar cells could underlie transient versus sustained excitatory input to ON-T and ON-S RGCs.

We next looked for ultrastructural correlates of differential synapse function in bipolar cell axon terminals reconstructed from SBFSEM volumes. Interestingly, type 6 bipolar cell synaptic ribbons were ~2- to 3-fold larger than those of type 5i bipolar cells (*Figure 9A, B*, *Figure 9—figure supplement 1A*). These differences did not depend on the identity of the postsynaptic RGC partner (*Figure 9D*, *Figure 9—figure supplement 1B*). Ribbons in XBCs, type 5o and 5t bipolar cells were also smaller than type 6 bipolar cell ribbons (*Figure 9D*, *Figure 9—figure supplement 1B*). This bipolar cell

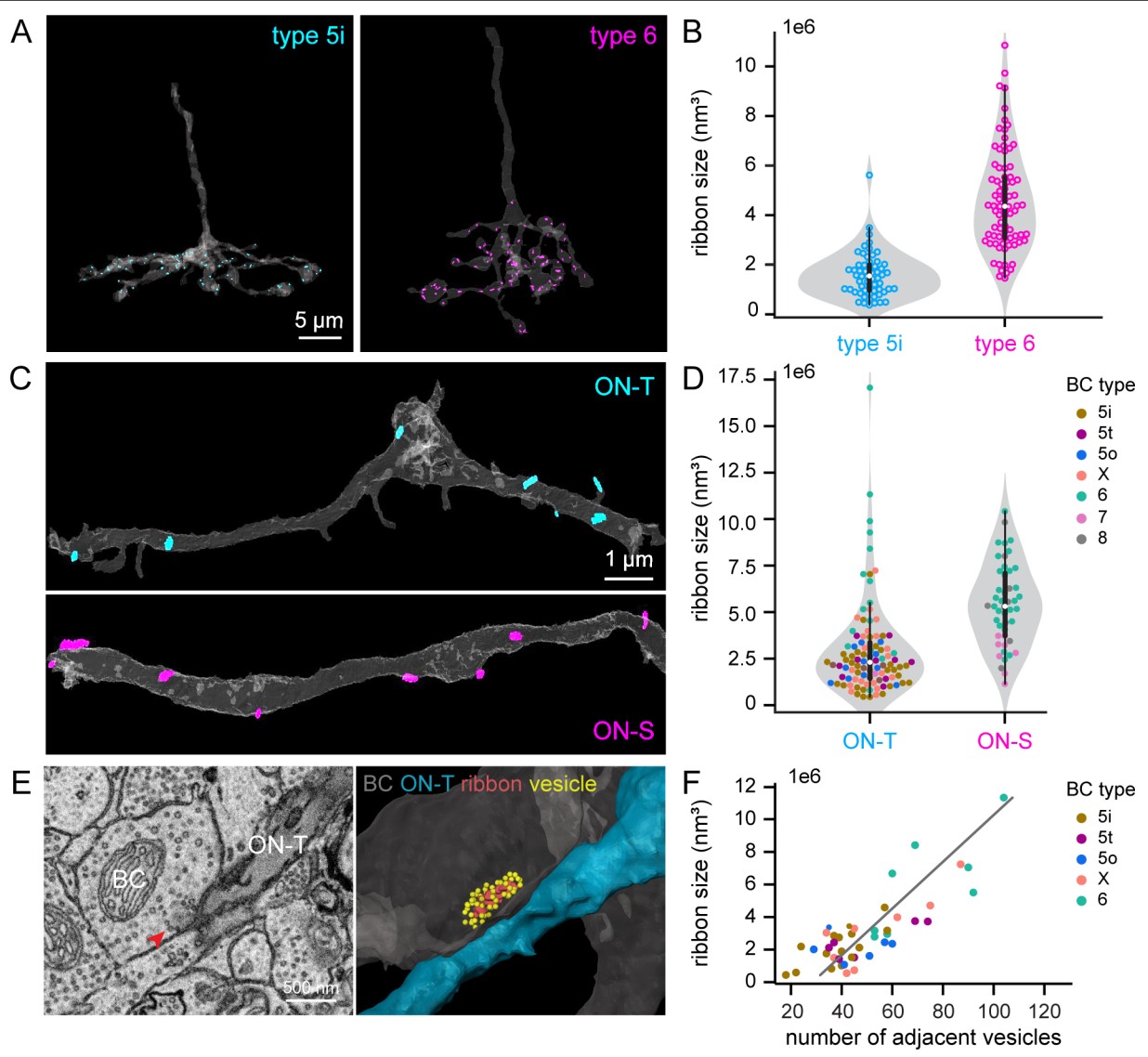

**Figure 9.** Bipolar cell subtype-specific differences in ribbon size (volume). (**A**) 3D reconstructions showing segmented axon stalk and axonal terminal (gray) and all the synaptic ribbons within the axonal terminal for a type 5i (cyan) and a type 6 (magenta) BC. (**B**) Violin plot showing distribution of ribbon volumes (each data point is a ribbon) for the type 5i and 6 BC in (**A**). The boxes indicate the interquartile range (thick lines, 25–75%), median value (white dot), and 1.5 times interquartile range (thin line). Shaded gray areas represent rotated kernel density plots. (**C**) 3D reconstructions showing segmented dendritic arbor (gray) and all the synaptic ribbons presynaptic to these segments of ON-T (cyan) or ON-S (magenta) retinal ganglion cell (RGC) dendrites. (**D**) Violin plot with scattered dots showing ribbon volumes for each ribbon presynaptic to a stretch of reconstructed ON-T or ON-S RGC dendrite. The color of each ribbon (dot) indicates the associated BC type. (**E**) An EM micrograph (left) and 3D reconstruction (right) of a ribbon and its adjacent vesicles. The ribbon is indicated by the red arrow in the EM image. In the 3D reconstruction, yellow dots indicate the adjacent vesicles surrounding the presynaptic ribbon (red) in the BC terminals (gray). Postsynaptic ON-T RGC is in blue. (**F**) Scatter plot showing ribbon size in volume in relationship with the number of adjacent vesicles. The gray line represents the linear regression model fitted to the data. Linear regression: $\beta$ = 6.98e−06, p < 0.001, $r^2$ = 0.66.

The online version of this article includes the following figure supplement(s) for figure 9:

**Figure supplement 1.** Bipolar cell subtype-specific differences in ribbon surface areas.

(**A**) Violin plot showing distribution of ribbon surface areas (each data point is a ribbon) for the type 5i and 6 BC in *Figure 9A*. (**B**) Violin plot with scattered dots showing ribbon surface areas for each ribbon presynaptic to a stretch of reconstructed ON-T or ON-S retinal ganglion cell (RGC) dendrite. (**C**) Scatter plot showing ribbon size in surface area in relationship with the number of adjacent vesicles. The gray line represents the linear regression model fitted to the data. Linear regression: $\beta$ = 0.0003, p < 0.001, $r^2$ = 0.69.

subtype-specific difference in ribbon size and the difference in ratio of type 6 to 5i input between RGC types meant that ribbons were on average smaller at synapses made on ON-T compared to ON-S dendrites (*Figure 9C, D*, *Figure 9—figure supplement 1B*).

## Discussion

We sought to understand how kinetically distinct visual pathways are established in the retina by directly comparing the visual response properties of transient and sustained ON RGCs with their presynaptic bipolar cells. Our results show that bipolar cell synapses have a key role in establishing different temporal channels in ON circuits of the retina. Further, amacrine cell-mediated presynaptic inhibition does not appear to account for transient versus sustained bipolar cell synaptic output. Rather, bipolar cell subtype-specific differences in synaptic release properties likely generate kinetically distinct output to postsynaptic targets. Our anatomical analysis suggests that these differences may at least in part be due to differences in synaptic vesicle pool size.

A key aspect of our work is that we measured RGC and bipolar cell responses using identical stimuli and under the same experimental recording conditions. To our knowledge, a direct comparison of RGC responses with those of their presynaptic bipolar cells has not previously been reported. The importance of measuring RGC and bipolar cell responses under the same conditions is highlighted by our observation that excitatory synaptic currents measured in ON alpha RGCs exhibited quantitatively different kinetics, although relative differences between types were preserved, when measurements were made from retinas maintained in ACSF versus Ames' bath solution (compare *Figure 2* (Ames') and *Figure 5*, *Figure 5—figure supplement 1* (ACSF)).

We found that the bipolar cell subtypes that provide the major input to ON-T (type 5i) or ON-S RGCs (types 6 and 7) had indistinguishable light-evoked voltage responses (*Figure 4*). This was not entirely surprising given that most or all ON cone bipolar cell subtypes are indirectly coupled to other nearby ON cone bipolar cells via electrical synapses between bipolar cell axon terminals and the distal processes of AII amacrine cells. These electrical synapses form the basis for the primary rod pathway that mediates vision under dim light conditions, but lateral signals can also flow among neighboring bipolar cells under conditions where cone signaling dominates (*Demb and Singer, 2012*). Indeed, a substantial fraction (~30–50%) of light-evoked responses in the bipolar cell types we investigated here can be attributed to lateral signals that arrive via electrical synapses at photopic light levels (*Kuo et al., 2016*). Each AII amacrine cell is electrically coupled to multiple ON cone bipolar subtypes as well as to other AII amacrine cells (*Marc et al., 2014*; *Tsukamoto and Omi, 2017*; *Veruki and Hartveit, 2002b*; *Veruki and Hartveit, 2002a*). Different ON cone bipolar cell subtypes may also be directly coupled via gap junctions to each other (*Sigulinsky et al., 2020*; *Tsukamoto and Omi, 2017*).

Direct or indirect (via AII) coupling may limit the extent to which different bipolar cells can generate subtype-specific voltage responses. These findings differ from a previous study that found some differences among mouse ON cone bipolar cell subtypes. *Ichinose et al., 2014* reported that a subset of type 5 bipolar cells, which the authors termed type 5f, and type 7 bipolar cells had more transient responses than type 6 bipolar cells. Methodological differences may account for the discrepancy between our results and the previous work by *Ichinose et al., 2014*. For example, Ichinose et al. included inhibitory receptor antagonists in their experimental solutions and used 100 µm diameter spots to characterize bipolar cell responses, whereas we left inhibitory signaling intact and used larger (520 µm diameter) spots to match the stimuli we used for RGC recordings. Interestingly, a recent investigation of spatial and temporal response properties of light-evoked intracellular calcium signals across bipolar cell types suggests that ON bipolar cells can exhibit type-specific response kinetics for small stimuli (<~200 µm diameter) but not larger spot sizes, presumably due to surround inhibition (*Hsiang et al., 2024*). Regardless of the reasons for the different findings, our observation that bipolar cell subtypes exhibited similar responses under the same recording and light stimulus conditions for which we observed obvious differences among postsynaptic RGCs argues that mechanisms upstream of bipolar cell glutamate release (e.g. cone-to-bipolar cell synapses or intrinsic voltage response properties of bipolar cells) are unlikely to have a major role in shaping RGC type-specific responses for the stimuli we used here.

A limitation of our work is that we did not record from every one of the bipolar cell subtypes that are presynaptic to ON-T RGCs. Instead, we measured responses from type 5i and 6 bipolar cells, which together comprise >~50% of the synaptic input to ON-T RGC dendrites. It is possible that

the other presynaptic bipolar cell types (type 5o, 5t, and XBC) may have more transient light-evoked voltage responses that could contribute to the transient excitatory synaptic input to ON-T RGCs. In fact, a previous study in mouse retina found that XBCs and a subset of type 5 bipolar cells express voltage-gated $Na^+$ channels (*Hellmer et al., 2016*). These channels may enhance response transience or might even contribute to the generation of light-driven all-or-none spiking responses in these bipolar cells (*Ichinose et al., 2014*; *Puthussery et al., 2013*; *Saszik and DeVries, 2012*).

Even if transient intrinsic response properties of type 5o, 5t, and XBC bipolar cells contribute to the generation of transient responses in ON-T RGCs, this alone is unlikely to account for the kinetics of the excitatory synaptic currents we measured in these RGCs. Type 5i and 6 bipolar cells have a significant sustained component of their light step-evoked voltage response (*Figure 4E, F*), whereas excitatory currents measured for the same stimulus and under the same conditions in ON-T RGCs had little to no sustained component (*Figure 2D, E*). We would expect to observe some sustained excitatory input to ON-T RGCs if type 5i and 6 bipolar cell synaptic output follows the voltage response of these bipolar cells. Thus, a marked transformation of the sustained bipolar cell membrane voltage response to a transient synaptic current must take place. Our measurements of extracellular glutamate at ON-T and ON-S dendrites (*Figure 5*) suggest that much of this transformation occurs at the level of bipolar cell glutamate release. It is interesting in this context that ON-T and ON-S RGCs both receive input from type 6 bipolar cells (comprising ~20% and ~50% of total synaptic input, respectively). Due to differences in dendrite stratification, type 6 bipolar cell synapses onto ON-T RGCs arise from a region of the axon that is located more proximal to the bipolar cell soma than the synapses formed onto ON-S RGCs, which are found within the distal axon lobules. However, we did not observe any differences in ribbon size at distal versus proximal locations in type 6 bipolar cell axons (*Figure 9D*). Whether postsynaptic target-specific synapse function, as has previously been reported in salamander retinal bipolar cells (*Asari and Meister, 2014*; *Asari and Meister, 2012*) and mouse starburst amacrine cells (*Pottackal et al., 2021*), contributes to signal diversification in circuits involving type 6 bipolar cells will require further investigation.

Our electrophysiological and glutamate imaging results indicate that amacrine cell-mediated inhibition at bipolar cell synaptic terminals is not necessary for generating transient glutamate release kinetics onto ON-T RGC dendrites. This is seemingly at odds with prior work that has indicated an essential role for surround inhibition in establishing type-specific bipolar cell glutamate release dynamics (*Franke et al., 2017*). However, we note that our analysis here focused only on whether presynaptic inhibition shapes the kinetics of excitatory inputs to ON-T RGCs. Our anatomical and pharmacological experiments together argue that presynaptic inhibition is not an important factor in the transformation from sustained bipolar cell voltage responses to transient excitatory synaptic currents in ON-T RGCs. It can be difficult to interpret the effects of bath-applied antagonists on RGC responses since there are multiple upstream pre- and postsynaptic targets at which these agents act within the retinal circuit. Pharmacological block of inhibitory receptors (*Figure 5*) not only alters cone bipolar cell responses, but also inhibitory networks among amacrine cells, and NBQX application (*Figure 7*) affects horizontal cell function and OFF bipolar cell circuits in addition to most amacrine cells. Despite these limitations, our results support a limited role for inhibition in generating transient excitatory input to ON-T RGCs. This is not to say that amacrine cell-mediated inhibition does not have a role in generating subtype-specific response properties for other aspects of bipolar cell function. Indeed, inhibition does have an important role in establishing differential spatial tuning of excitatory inputs to the RGC types we studied; excitatory synaptic currents in ON-T RGCs exhibit stronger surround inhibition than ON-S RGCs, and this surround inhibition is reduced by bath application of GABA receptor antagonists (*Figure 5—figure supplement 1C*). Interestingly, recent work revealed postsynaptic target-specific surround inhibition at neighboring type 6 bipolar cell output synapses within the same axon terminal (*Swygart et al., 2024*). Whether target-specific presynaptic inhibition of type 6 bipolar cells also contributes to differential surround inhibition between ON-T and ON-S RGCs or is primarily due to differences across bipolar cell types (e.g. type 5i/5t/XBC vs. type 6/7) remains to be determined.

We conclude that transient versus sustained excitatory inputs to ON-T and ON-S RGCs most likely arise from bipolar cell subtype-specific differences in synaptic release properties. Excitatory inputs to ON-T RGCs exhibited stronger paired-flash depression than those to ON-S RGCs (*Figure 8*), which indicates distinct vesicle cycling dynamics at the bipolar cell axon terminals presynaptic to the different

RGC types. Our ultrastructural analysis suggests that bipolar subtype-dependent differences in ribbon size may be a factor in establishing transient versus sustained glutamate release kinetics (*Figure 9*). It is important to note, however, that type 7 bipolar cells (~45% of excitatory input to ON-S RGCs) have small synaptic ribbons, yet iGluSnFR imaging of type 7 bipolar cell axon terminals found that these cells exhibit sustained glutamate release (*Srivastava et al., 2022*).

We found that ribbon size is strongly correlated with the total number of vesicles that are in close contact with the synaptic ribbon across different ON bipolar cell types (*Figure 9E, F*, *Figure 9—figure supplement 1C*). Thus, there are fewer ribbon-associated vesicles at glutamatergic synapses formed on ON-T RGC dendrites compared to those at ON-S RGCs. Ribbon-associated vesicles are thought to correspond to the releasable pool of vesicles, comprised of both a readily releasable pool that is docked at the base of the ribbon, at the presynaptic active zone, and a reserve (alternatively referred to as 'releaseable') pool of vesicles that are bound to the ribbon at more distal sites (*Grabner et al., 2023*; *Lagnado and Schmitz, 2015*; *von Gersdorff et al., 1996*; *Wan and Heidelberger, 2011*). The number of ribbon-associated vesicles, as well as the rate at which the readily releasable and reserve vesicle pools are replenished following exocytosis, together define the amplitude and time course of stimulus-evoked vesicle release (*Jackman et al., 2009*; *Lagnado and Schmitz, 2015*; *Neves and Lagnado, 1999*; *Oesch and Diamond, 2011*; *Singer and Diamond, 2006*; *von Gersdorff and Matthews, 1997*). For example, a small pool of readily releasable vesicles and relatively slow replenishment in rod bipolar cell axon terminals contributes to transient release kinetics from these cells (*Singer and Diamond, 2006*). Our observation that the average excitatory synaptic input to ON-T RGCs is smaller and that most excitatory input to ON-T cells originates from bipolar synapses with fewer ribbon-associated vesicles is consistent with a role for vesicle depletion in establishing transient glutamate release onto these RGCs. The larger pool of ribbon-associated vesicles in type 6 bipolar cell terminals, which comprise ~55% of the synaptic input to ON-S RGCs, may support sustained synaptic transmission onto these RGCs. However, additional mechanisms, such as bipolar cell type-specific differences in the rate of refilling of the ribbon-associated vesicles, could also contribute to sustained versus transient glutamate release.

Direct measurements of exocytosis at ribbon synapses show that release rates increase with increasing ribbon size and the number of vesicles bound to the ribbon. Evidence for this dependence includes a tight correspondence between functional measures of the releasable vesicle pool and anatomical measurements of the number of vesicles bound to the ribbon (*Grabner et al., 2023*; *von Gersdorff et al., 1996*). Ribbon size and the number of ribbon-associated vesicles vary across the day–night cycle; these morphological changes are correlated with diurnal changes in maintained vesicle fusion rates (*Babai et al., 2016*; *Dembla et al., 2020*; *Hull et al., 2006*). Finally, release rates decrease when ribbon size is reduced (*Mehta et al., 2013*; *Wakeham et al., 2023*), or when ribbons are ablated either abruptly or genetically (*Grabner and Moser, 2021*; *Maxeiner et al., 2016*; *Mehta et al., 2013*; *Snellman et al., 2011*). These biophysical studies are consistent with the correlation that we observe here between ribbon size and the extent to which transmitter release is transient versus sustained. A note of caution in making this comparison is that exocytosis is sensitive to the voltage trajectory (*Snellman et al., 2009*), and the biophysical studies often use more rapid and larger voltage changes than those elicited by physiological light inputs. In fact, genetic deletion of ribbons does not strongly alter light-evoked excitatory currents in ON-S RGCs (*Okawa et al., 2019*). Why ribbon deletion has relatively little effect on light-evoked responses in ON-S RGCs remains to be determined. It is interesting to note that ~30% of synapses formed by type 6 bipolar cells onto ON-S RGC dendrites lack ribbons even in wild-type retina (*Okawa et al., 2019*). Future studies will be needed to investigate to what extent such ribbonless synapses could contribute to bipolar cell type-specific differences in glutamate release such as we observed here.

The dynamics of transmitter release are also impacted by vesicle depletion and the time required for refilling of releasable vesicle pools following strong stimulation. In particular, the sustained transmitter release rate should be equal to the sustained refilling rate. Direct measurements of refilling at the cone output synapse following strong activation show that it is quite slow – with a time constant near 1 s (*Grabner et al., 2016*; *Jackman et al., 2009*; *Rabl et al., 2006*). This is far too slow to explain the recovery from paired-flash depression that we observe here. Several factors may contribute to this difference. First, postsynaptic receptor saturation can contribute to paired-pulse depression, and it can recover from such depression more quickly than the recovery of presynaptic release (*Grabner*

et al., 2016). Second, the paired-flash depression that we measure in RGC synaptic inputs is largely absent in the bipolar voltage responses. This suggests that it is dominated by a history dependence at the cone bipolar output synapses and refilling of vesicle pools may proceed more quickly at bipolar synapses than cone synapses.

Mechanisms that we did not directly investigate here may also contribute to the difference between transient and sustained responses. For example, presynaptic resting potential has a strong effect on steady-state vesicle release and thus stimulus-evoked release dynamics. Exocytosis that occurs when bipolar cells rest at voltages near release threshold can lead to a depletion of readily releasable vesicles that limits the amount of release that occurs upon further light-induced depolarizations (*Grimes et al., 2014*; *Jarsky et al., 2011*; *Oesch and Diamond, 2011*). Thus, an initial phasic component of release can be reduced by depolarized resting potentials, whereas hyperpolarized potentials can facilitate greater occupancy of vesicles in the releasable pool, which can enhance response transience. Bipolar subtype-dependent differences in release threshold due to differential $Ca^{2+}$ channel expression/regulation (*Pangrsic et al., 2018*) and/or coupling between $Ca^{2+}$ channel activity and vesicle release (*Burrone et al., 2002*; *Burrone and Lagnado, 2000*; *Ghosh et al., 2004*) could also impact release dynamics. Several aspects of excitatory synaptic currents in the two RGC types we studied here are consistent with a situation in which bipolar cell inputs to ON-T RGCs have a higher threshold for glutamate release compared to those at ON-S RGCs; ON-T RGCs receive less spontaneous excitatory synaptic input than ON-S RGCs (*Kuo et al., 2016*) and stimulus-evoked EPSCs are slower, sparser, and more biased toward larger amplitude events in ON-T RGCs compared to those in ON-S RGCs (*Figure 2*, *Figure 1—figure supplement 1B*). Further investigation will be required to determine whether this is due to differences in axon terminal resting potential and/or other presynaptic factors. We did not find evidence for subtype-specific differences in resting membrane potential among type 5i, 6, and 7 bipolar cells, but our somatic voltage recordings likely have limited ability to resolve small voltage differences at the bipolar cell axon terminals. Further work is also necessary to determine to what extent molecular differences in presynaptic release machinery may contribute to bipolar subtype-specific glutamate release dynamics (*Thoreson and Zenisek, 2024*).

Our results do not rule out additional roles for postsynaptic mechanisms in establishing transient versus sustained RGC spike output. For example, delayed amacrine cell-mediated inhibitory synaptic input to RGC dendrites could truncate spiking responses (*Nirenberg and Meister, 1997*). We did not observe a significant difference in the light-evoked excitation/inhibition amplitude ratio between ON-S and ON-T RGCs (*Figure 2—figure supplement 2E*), but our data does not exclude a role for postsynaptic inhibition in generating transient responses. RGC cell-intrinsic mechanisms, such as RGC subtype-specific differences in voltage-gated $Na^+$ channel properties and/or expression pattern could also contribute to distinct response kinetics (*Brombas et al., 2022*; *Chang et al., 2023*; *Werginz et al., 2020*). While a combination of pre- and postsynaptic factors may ultimately act in concert to generate transient versus sustained RGC spike output (*Warwick et al., 2018*; *Werginz et al., 2020*), our results establish a fundamental role for bipolar cell synapses in the segregation of visual input into kinetically distinct channels in ON pathways of the retina.

## Materials and methods

### Animals

Experiments used 5- to 15-week-old wild-type, *Grm6-tdTomato* (*Morgan et al., 2011*), *Gus8.4-EGFP* (*Huang et al., 1999*; JAX 026704), *Gjd2-EGFP*, or *Kcng4-Cre* (*Duan et al., 2014*; JAX 029414) transgenic mice. Transgenic mouse strains were backcrossed into the same genetic background as wild-type mice (C57BL/6J; Jackson Laboratory). Animal care and handling followed procedures approved by the Institutional Animal Care and Use Committee of the University of Washington or the University of Victoria Animal Care Committee.

### Tissue preparation

Retinal tissue was prepared for physiological recordings as previously described (*Kuo et al., 2016*; *Srivastava et al., 2022*). Briefly, mice were killed by cervical dislocation or decapitation and retinas were dissected from both eyes in oxygenated (95% $O_2$ and 5% $CO_2$), bicarbonate-buffered Ames' (Sigma) or ACSF solution and either flat-mounted with the vitreal surface facing up onto a

poly-D-lysine coated glass coverslip (Corning) (RGC patch-clamp recordings and iGluSnFR imaging) or sliced in an orientation perpendicular to the flat-mount plane using a vibratome (200 µm slice thickness; bipolar cell recordings) before being transferred to an upright 2P fluorescence microscope. ACSF contained (in mM): 125 NaCl, 2.5 KCl, 1.2 $CaCl_2$, 1.5 $MgCl_2$, 1.25 $NaH_2PO_4$, 20 glucose, 0.5 L-glutamine, 0.1 sodium ascorbate, and 26 $NaHCO_3$ (pH 7.4, 310 mOsm/l). For preparation of retinal slices, isolated retina was surrounded with melted agarose (maintained at ~40°C) followed by rapid cooling using an ice block. The embedded retina was then positioned in front of the vibratome blade in an orientation orthogonal to the flat-mount plane for sectioning. All experiments exclusively used tissue from ventral retina. All electrophysiological data show results from at least two different retinas.

## Electrophysiology

During recordings, retinal tissue was continuously perfused (~6–8 ml/min) with oxygenated, bicarbonate-buffered Ames' solution maintained at 30–34°C. In a subset of electrophysiological recordings (*Figure 6C–F*, *Figure 5—figure supplement 1*), tissue was perfused with ACSF rather than Ames' solution. Electrical signals were acquired at 10 kHz and low-pass filtered at 3 kHz using a Multiclamp 700B or Multiclamp 700A amplifier (Molecular Devices) and Symphony Data Acquisition software (http://symphony-das.github.io). The somas of different ON bipolar cell subtypes were targeted for gramicidin perforated-patch current-clamp recordings using 2P laser scanning excitation (960 nm) in retinal slices prepared from transgenic mouse lines in which bipolar cell subtypes expressed fluorescent proteins. Tissue from *Grm6-tdTomato* mice, in which a sparse subset of ON bipolar cell types (type 6, 7, and 8) express tdTomato, was used to target recordings to type 6 bipolar cells (*Morgan et al., 2011*; *Schwartz et al., 2012*). Type 7 bipolar cells were targeted for recording in *Gus8.4-EGFP* mice, which express EGFP in type 7 bipolar cells and rod bipolar cells (*Huang et al., 2003*). Type 5i bipolar cells were targeted in retinas from *Gjd2-EGFP* mice, which we found express EGFP selectively in these bipolar cells (*Figure 3—figure supplement 2*) as well as less robustly in an uncharacterized amacrine cell type (not shown). For bipolar cell recordings, we minimized light exposure from the 2P laser by restricting scan regions to the inner retina, away from photoreceptors, and by keeping laser exposure brief (<~30 s/scan region; <2 mW post-objective laser power) (*Kuo et al., 2016*). After identification of fluorescent cell bodies, we acquired recordings using infrared differential interference contrast microscopy. Bipolar recordings used an internal solution containing 123 mM K-aspartate, 10 mM KCl, 10 mM HEPES, 1 mM $MgCl_2$, 1 mM $CaCl_2$, 2 mM EGTA, 4 mM Mg-ATP, 0.5 mM Tris-GTP, 100 µM Alexa Fluor 594-hydrazide, and 25–30 µg/ml gramicidin D (280 mOsm/l; pH 7.2). Pipettes (~7–9 Ω tip resistance) were front-filled with gramicidin-free internal solution to facilitate formation of high resistance (>1 GΩ) seals between patch pipettes and the cell membrane. After seal formation, series resistance ($R_s$) was monitored in voltage-clamp mode by applying a 10 mV voltage step across the pipette tip. Experiments were initiated once $R_s$ declined to <50 MΩ, which typically took 20–30 min. During recordings, we occasionally confirmed the integrity of the cell membrane under the recording electrode by monitoring Alexa Fluor 594 fluorescence using 2P excitation; if any dye was visible in the cell body, we abandoned the recording. No holding current was applied during current-clamp recordings from bipolar cells. At the conclusion of each bipolar cell recording, the membrane under the patch pipette was intentionally ruptured and fluorescence images of Alexa Fluor 594-filled cells were acquired using 2P microscopy to confirm bipolar cell subtype identities (*Wässle et al., 2009*).

RGC recordings were performed in a flat-mount preparation of the retina from wild-type mice or *Gjd2-EGFP* transgenic mice following previously described methods (*Kuo et al., 2016*). Excitatory synaptic currents were isolated in whole-cell voltage-clamp recordings by holding RGCs near the reversal potential for Cl⁻-mediated conductances (–68.5 mV). Patch pipettes (2.5–4 MΩ) were filled with an internal solution containing (in mM): 105 Cs-methanesulfonate, 10 Tetraethylammonium-Cl, 20 HEPES, 10 EGTA, 2 QX-314, 5 Mg-ATP, and 0.5 Tris-GTP (~280 mOsm; pH 7.2). Series resistance (<15 MΩ) was compensated by 70% using the amplifier circuitry. Inhibitory receptor antagonists (SR-95531, TPMPA, strychnine) were obtained from Tocris or Sigma-Aldrich and added to the bath solution from concentrated aqueous stock solutions.

## iGluSnFR expression and two-photon imaging

Intravitreal injection of a cre-dependent viral vector (pAAV.hSyn.Flex.iGluSnFR.WPRE.SV40; Addgene Catalog #98929-AAV1) was performed into Kcng4-cre mice to selectively express the glutamate sensor, iGluSnFR, in alpha RGCs, including ON-T and ON-S RGCs (*Krieger et al., 2017*). All injection procedures, tissue preparations, and subsequent two-photon imaging were performed as previously described in *Srivastava et al., 2022*. Briefly, for iGluSnFR imaging, the laser was tuned to 920 nm and all recordings were acquired at a frame rate of 58.25 Hz (256 × 256 pixels). In KCNG4-Cre mice, ON-S alpha RGCs were first identified by their spiking properties using extracellular cell-attached recordings. To identify the dendrites of ON-S RGCs, cells were filled with sulforhodamine 101. ON-T RGC dendrites were then recorded at the same location but deeper into the retina (*Krieger et al., 2017*).

## Visual stimulation

Visual stimuli were presented from a photopic mean background illumination (~3300 rhodopsin isomerizations per rod per second (R*/rod/s)) to focus specifically on cone-mediated signaling (*Grimes et al., 2018*). Visual stimuli for electrophysiological recordings were generated using Stage Visual Stimulation Software (http://stage-vss.github.io). For all electrophysiological recordings except those in *Figure 6C–F*, *Figure 5—figure supplement 1*, light stimuli consisted of a spatially uniform circular spot (520 μm diameter) delivered from an LED (405 nm peak output; Hosfelt) that was focused through the microscope substage condenser (*Kuo et al., 2016*). In the experiments shown in *Figure 6C–F*, *Figure 5—figure supplement 1*, a Digital Light Processing (DLP) projector driven by a 405-nm LED (Wintech Digital) was used to project spots of various sizes through the microscope objective (*Kuo et al., 2020*). For iGluSnFR imaging (*Figure 5*), background illuminance was kept around 1000 photons/μm$^2$/s. For experiments, static spots of different sizes were generated and presented using StimGen software (https://github.com/benmurphybaum/StimGen, copy archived at *Murphy-Baum, 2019*). Stimuli were presented via a DLP through the substage condenser focused onto the photoreceptor layer of the retina. All visual stimuli were presented 4 s after the start of the recording for a duration of 2 s.

## Data analysis and statistics

STA waveforms (*Figure 1*) and LN models of light-driven RGC (*Figure 2*) and bipolar cell (*Figure 4*) responses were derived from responses to randomly fluctuating light stimuli (Gaussian distribution of light intensities with standard deviation = 50% of mean intensity; 0–60 Hz bandwidth) as described previously (*Chichilnisky, 2001*; *Kim and Rieke, 2001*; *Rieke, 2001*). For LN model construction, the linear filter was calculated by computing the cross-correlation of the light stimulus with the measured voltage (bipolar cell) or current (RGC) response and dividing the result by the power spectrum of the stimulus. The static nonlinearity was constructed from a point-by-point comparison of the predicted filter output with the measured response and binning the result along the prediction axis. In almost all cases, the linear filter had a biphasic shape with initial and secondary lobes that are opposite polarity. A small number of bipolar cells had triphasic filters (*Figure 4—figure supplement 1*). We quantified the kinetics of each STA or linear filter by measuring the time at which the STA/filter waveform crossed zero ('zero-cross time'; e.g. *Figure 1C*, left) between the first and second lobes of the filter (e.g. between peak and trough of STA waveforms) and by calculating a biphasic index = $|A|/|B|$, where $|B|$ is the absolute peak amplitude of the initial lobe of the filter and $|A|$ is the amplitude of the peak amplitude of the secondary lobe (e.g. *Figure 2C*, right). We quantified the kinetics of responses to light step stimuli by calculating the time between light step onset and the peak amplitude response ('time to peak'; e.g. *Figure 2E*, left) and dividing the mean steady-state response (calculated from the last 100-ms light step increment) by the peak amplitude initial response at step onset (and multiplying by 100) (e.g. *Figure 2E*, right). We note that alternative measures of step response or linear filter kinetics than those presented in *Figure 4* were also not statistically different between bipolar cell subtypes (not shown). For light step-evoked responses, steady-state measurements were obtained from the mean current/voltage over the last 100 ms of the 500-ms light steps (electrophysiology) or the mean fluorescence over the last 1 s of the 2 s light step (iGluSnFR). Statistical tests were performed using MATLAB (Mathworks) or Igor Pro (WaveMetrics) software.

## Biolistic transfection

DNA plasmids encoding mTFP1-myc (24 µg) and PSD95-mCherry (12 µg) under the control of the cytomegalovirus promoter were coated onto gold particles (1.6 µm diameter, 12.5 mg, Bio-Rad). Using a Helios gene gun (40 psi, Bio-Rad), these particles were then delivered to RGCs in whole-mount retinas. Subsequently, the transfected retinas were incubated overnight in mACSF within a humid, oxygenated chamber maintained at 33°C.

## Immunohistochemistry

Retinal sections (60 µm thick) were prepared using a vibratome (Leica) from retinal whole-mount retinas of *Gjd2-EGFP* mice. Both whole-mount retinas and retinal sections were fixed in 4% PFA for 30 min. Retinal sections were embedded in low melting point agarose (Sigma-Aldrich). Retinal tissues were first incubated with blocking solution (5% normal donkey serum, 0.5% Triton X-100 in PBS, Sigma-Aldrich) for 2 hr at 4°C, then incubated overnight at 4°C with primary antibodies dissolved in blocking solution. Excess primary antibody was rinsed 3 × 10 min with PBS, then sections were incubated for 2 hr with secondary antibodies dissolved in PBS. Excess secondary antibody was rinsed 3 × 10 min with PBS and sections were mounted on standard microscopy glass slides using Vectashield mounting medium (Vector). Primary antibodies used in the study were goat anti-ChAT diluted 1:1000 Chemicon AB144P, mouse anti-CaBP5 diluted 1:1000 courtesy of Dr. Haeseleer, mouse anti-myc diluted at 1:1000, and chicken anti-GFP diluted at 1:1000. Secondary antibodies were donkey anti-goat IgG Dylight-568 diluted 1:1000, donkey anti-mouse Dylight-647 diluted 1:1000, donkey anti-mouse Dylight-405 diluted at 1:1000, donkey anti-chicken Alexa 488, and the nuclear dye TOPRO3 (diluted 1:200, Thermo Fisher).

## NIRB and SBFSEM

Two neighboring ON-T RGCs were physiologically identified in *Gjd2-EGFP* mice. After identifying ON-T RGCs via their characteristic step-evoked extracellular spike response (e.g. *Figure 1B*), they were filled with Alexa 488 or Alexa 555. Dendritic arbors and T5i-BC axonal arbors were imaged using a confocal microscope. The retina was then fixed with 4% glutaraldehyde for 30 min at room temperature (RT). The retina was washed with 0.1 M Na$^+$ Cacodylate buffer three times and flat mounted with 0.1 M Na$^+$ Cacodylate buffer. To relocate the two RGCs under electron microscopy, we applied the NIRB technique previously published to burn fiducial markers in the retina (Bishop et al.). After NIRBing, the retina tissue was unmounted and fixed in 4% glutaraldehyde overnight. The retina was processed for SBFSEM subsequently according to the protocol described previously (*Della Santina et al., 2016*). Briefly, the tissue was initially treated with reduced osmium for 1 hr in the refrigerator. Subsequently, after washing with double-distilled H$_2$O (ddH$_2$O), the tissue was placed in a freshly prepared thiocarbohydrazide solution for 20 min at RT. Following another round of ddH$_2$O rinsing, the tissue was incubated in 2% osmium tetroxide for 30 min at RT. After rinsing, the sample was incubated in 1% uranyl acetate at 4°C overnight. Post-wash, the tissue was stained with Walton's lead aspartate for 30 min at 60°C. Following another wash, the retinal piece underwent dehydration using a graded alcohol series (20%, 50%, 70%, 90%, 100%, and 100%), each step lasting 5 min, followed by two changes of 100% propylene oxide at RT for 10 min each. Finally, the tissue was embedded in Epon. The resulting block was then trimmed and mounted in the SBEM microscope (GATAN/Zeiss Sigma, 3View), and image stacks were acquired at a voxel size of 5 × 5 × 50 nm.

## Additional information

### Competing interests

Fred Rieke: Reviewing editor, eLife. The other authors declare that no competing interests exist.

### Funding

| Funder | Grant reference number | Author |
|---|---|---|
| National Eye Institute | EY-033932 | Sidney P Kuo |

| Funder | Grant reference number | Author |
| --- | --- | --- |
| National Eye Institute | EY10699 | Rachel OL Wong |
| National Eye Institute | EY028542 | Fred Rieke |

The funders had no role in study design, data collection, and interpretation, or the decision to submit the work for publication.

## Author contributions

Sidney P Kuo, Wan-Qing Yu, Conceptualization, Data curation, Formal analysis, Investigation, Methodology, Writing – original draft, Writing – review and editing; Prerna Srivastava, Data curation, Formal analysis, Investigation; Haruhisa Okawa, Luca Della Santina, David M Berson, Data curation, Formal analysis, Investigation, Methodology; Gautam B Awatramani, Data curation, Formal analysis, Investigation, Methodology, Writing – review and editing; Rachel OL Wong, Fred Rieke, Conceptualization, Data curation, Formal analysis, Funding acquisition, Investigation, Methodology, Writing – original draft, Project administration, Writing – review and editing

## Author ORCIDs

Luca Della Santina ⓘ http://orcid.org/0000-0001-6036-1295
Gautam B Awatramani ⓘ https://orcid.org/0000-0002-0610-5271
Rachel OL Wong ⓘ https://orcid.org/0000-0001-5089-6296
Fred Rieke ⓘ https://orcid.org/0000-0002-1052-2609

## Ethics

Animal care and handling followed procedures approved by the Institutional Animal Care and Use Committee of the University of Washington (protocol 3030-01) and the University of Victoria Animal Care Committee (protocol AE-24-012-01).

Reviewer #1 (Public review): https://doi.org/10.7554/eLife.98817.3.sa1
Reviewer #2 (Public review): https://doi.org/10.7554/eLife.98817.3.sa2
Reviewer #3 (Public review): https://doi.org/10.7554/eLife.98817.3.sa3
Author response https://doi.org/10.7554/eLife.98817.3.sa4

# Additional files

## Supplementary files

MDAR checklist

## Data availability

Source data for Figures 1–5, 8 and 9 have been deposited with Dryad: Cone bipolar cell synapses generate transient versus sustained signals in parallel ON pathways of the mouse retina, Rieke, Kuo, Yu, Srivastava, Okawa, Della Santina, Berson, Awatramani, Wong, 2025; Dryad. https://doi.org/10.5061/dryad.47d7wm3tz.

The following dataset was generated:

| Author(s) | Year | Dataset title | Dataset URL | Database and Identifier |
| --- | --- | --- | --- | --- |
| Kuo SP, Yu WQ, Srivastava P, Okawa H, Santina LD, Berson DM, Awatramani GB, Wong ROL, Rieke F | 2026 | Data from: Cone bipolar cell synapses generate transient versus sustained signals in parallel ON pathways of the mouse retina | https://doi.org/10.5061/dryad.47d7wm3tz | Dryad Digital Repository, 10.5061/dryad.47d7wm3tz |

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
