## [Editor Report · eLife Assessment]

This **important** study shows that retinal bipolar cell subtype-specific differences in the size of synaptic ribbon-associated vesicle pools contribute to the transient versus sustained kinetics of the responses of retinal ganglion cells. The data are extensive and **compelling**. This work will be of broad interest to researchers working on synaptic transmission, retinal signal processing, and sensory neurobiology.

---

## [Referee Report · Reviewer #1 (Public review)]

Summary:

In the retina, parallel processing of cone photoreceptor output under bright light conditions dissects critical features of our visual environment, and fundamental to visual function. Cone photoreceptor signals are sampled by several types of bipolar cells and passed onto the ganglion cells. At the output of retinal processing, retinal ganglion cells send about 40 different codes of the visual scene to the brain for further processing. In this study, the authors focus on whether subtype-specific differences in the size of synaptic ribbon-associated vesicle pools of bipolar cells contribute to different retinal ganglion cell (RGC) responses.

Specifically, inputs to ON alpha RGCs producing transient versus sustained kinetics (ON-S vs. ON-T, respectively) are compared. The authors first demonstrate that ON-S vs. ON-T RGCs are readily identifiable in a whole mount preparation and respond differently to both static and to a spatially uniform, randomly fluctuating (Gaussian noise) light stimulus. Liner-nonlinear (LN) models were used to estimate the transformation between visual input and excitatory synaptic input for each RGCs; these models suggested the presence of transient versus sustained kinetics already in the excitatory inputs to ON-T and ON-S RGCs.

Indeed, the authors show that (glutamatergic) excitatory inputs to ON-S vs. ON-T RGCs are of distinct kinetics. The subtypes of bipolar cells providing input to ON-S are known (i.e., type 6 and 7), but the source of excitatory bipolar inputs to ON-T RGCs needed to be determined. In a tedious process, it is elegantly shown here that ON-T RGCs receive most of their excitatory inputs from type 5 and 6 bipolars. Interestingly, the temporal properties of light-evoked responses of type 5, 6 and 7 bipolars recorded from the somas were indistinguishable and rather sustained, suggesting that the origin of transient kinetics of excitatory inputs to ON-T RGCs suggested by the LN model might be found in the processing of visual signals at the bipolar cell axon terminal. Blocking GABA- or glycinergic inhibitory inputs did not alter the light-evoked excitatory input kinetics to ON-T and ON-S RGCs. Two-photon glutamate sensor imaging revealed significantly faster kinetics of light-evoked glutamate signals at ON-T versus ON-S RGCs, and that differences in glutamate release from presynaptic bipolar cells are retained without amacrine feedback to bipolar cells. Detailed EM analysis of bipolar cell ribbon synapses onto ON-T and ON-S RGCs revealed fewer ribbon-associated vesicles at ON-T synapses, that is consistent with stronger paired-flash depression of light-evoked excitatory currents in ON-T RGCS versus ON-S RGCs. This study suggests that bipolar subtype-specific differences in the size of synaptic ribbon-associated vesicle pools contributes to transient versus sustained kinetics in RGCs.

Strengths:

The use of multiple, state-of-the-art tools and approaches to address the kinetics of bipolar to ganglion cell synapse in an identified circuit.

---

## [Referee Report · Reviewer #2 (Public review)]

Summary:

Goal of the study. The authors tried to pinpoint the origins of transient and sustained responses measured at retinal ganglion cells (rgcs), which is the output layer of the retina. Response characteristics of rgcs are used to group them into different types. The diversity of rgc types represents the ability of the retina to transform visual inputs into distinct output channels. They find that the physical dimensions of bipolar cell's synaptic ribbons (specialized release sites/active zones) vary across the different types of cone on-bpcs, in ways that they argue could facilitate transient or sustained release. This diversity of release output is what they argue underlies the differences in on-rgcs response characteristics, and ultimately represents a mechanism for creating parallel cone-driven channels.

Strengths:

The major strengths of the study are the anatomical approaches employed and the use of the "glutamate sniffer" to assay synaptic glutamate levels. The outline of the study is elegant and reflects the strengths of the authors.

Comments on revised version:

The authors have addressed my comments either through new experiments and/or with additional citations.

Explanation of the studies significance. I think the study provides a solid set of data, acquired through exceptional methodologies, and delivers a compelling hypothesis. This is an exceptionally talented group of systems level thinkers and experimentalists, who are now pointing to smaller scale biophysical principles of synaptic transmission.

---

## [Referee Report · Reviewer #3 (Public review)]

Summary:

Different types of retinal ganglion cell (RGC) have different temporal properties - most prominently a distinction between sustained vs. transient responses to contrast. This has been well established in multiple species, including mouse. In general, RGCs with dendrites that stratify close to the ganglion cell layer (GCL) are sustained; whereas those that stratify near the middle of the inner plexiform layer (IPL) are transient. This difference in RGC spiking responses aligns with similar differences in excitatory synaptic currents as well as with differences in glutamate release in the respective layers - shown previously and here, with a glutamate sensor (iGluSnFR) expressed in the RGCs of interest. Differences in glutamate release were not explained by differences in the distinct presynaptic bipolar cells' voltage responses, which were quite similar to one another. Rather, the difference in transient vs. sustained responses seems to emerge at the bipolar cell axon terminals in the form of glutamate release. This difference in the temporal pattern of glutamate release was correlated with differences in the size of synaptic ribbons (larger in the bipolar cells with more sustained responses), which also correlated with a greater number of vesicles in the vicinity of the larger ribbons.

The main conclusion of the study relates to a correlation (because it is difficult to manipulate ribbon size or vesicle density experimentally): the bipolar cells with increased ribbon size/vesicle number would have a greater possibility of sustained release, which would be reflected in the postsynaptic RGC synaptic currents and RGC firing rates. This model proposes a mechanism for temporal channels that is independent of synaptic inhibition. Indeed, some experiments in the paper suggest that inhibition cannot explain the transient nature of glutamate release onto one of the RGC types. Still, it is surprising that such a diverse set of inhibitory interneurons in the retina would not play some role in diversifying the temporal properties of RGC responses.

Strengths:

(1) The study uses a systematic approach to evaluating temporal properties of retinal ganglion cell (RGC) spiking outputs, excitatory synaptic inputs, presynaptic voltage responses, and presynaptic glutamate release. The combination of these experiments demonstrates an important step in the conversion from voltage to glutamate release in shaping response dynamics in RGCs.

(2) The study uses a combination of electrophysiology, two-photon imaging and scanning block face EM to build a quantitative and coherent story about specific retinal circuits and their functional properties.

Weaknesses:

(1) There were some interesting aspects of the study that were not completely resolved, and resolving some of these issues may go beyond the current study. For example, it was interesting that different extracellular media (Ames medium vs. ACSF) generated different degrees of transient vs. sustained responses in RGCs, but it was unclear how these media might have impacted ion channels at different levels of the circuit that could explain the effects on temporal tuning.

(2) It was surprising that inhibition played such a small role in generating temporal tuning. The authors explored this further in the revision, which supported the original claim that inhibition plays a minor role in glutamate release dynamics from the bipolar cells under study.

---

## [Author Response]

The following is the authors’ response to the current reviews.

Reviewer #2 had several remaining suggestions:

In some instances, the authors face well-known limitations. For example, bath application of drugs. Blockers of Gly and Gaba receptors are likely problematic when studying a network that includes a diverse set of inhibitory interneurons. Likewise, the results derived from application of AMPAR and KAR blockers should impact HC cell fxn, and presumably inner retina interneuron networks. In the Discussion the authors are encouraged to address more of these concerns (e.g., Discussion line 709).Rather than concluding that the bath application of drugs is without complications, they can conclude that under the experimental conditions, glutamate release from these On-bipolars continues to exhibit Transient and Sustained release. This is really the key point of their study.

This is a good suggestion. We have added a discussion of the complications of the pharmacology starting on line 754.

If indeed sustained release is a reflection of higher release rates, ribbon size is what point to but, there are many other possibilities, such as SV recycling, or recruitment of reserve pools of SVs, fusion machinery, Cav channel behavior. The authors could cite more literature in the Discussion.

We added a sentence to this effect in the discussion, starting on line 866.

The following is the authors’ response to the original reviews.

**Reviewer #1 (Public Review):**
Summary:In the retina, parallel processing of cone photoreceptor output under bright light conditions dissects critical features of our visual environment and is fundamental to visual function. Cone photoreceptor signals are sampled by several types of bipolar cells and passed onto the ganglion cells. At the output of retinal processing, retinal ganglion cells send about 40 different codes of the visual scene to the brain for further processing. In this study, the authors focus on whether subtype-specific differences in the size of synaptic ribbon-associated vesicle pools of bipolar cells contribute to different retinal ganglion cell (RGC) responses. Specifically, inputs to ON alpha RGCs producing transient versus sustained kinetics (ON-S vs. ON-T, respectively) are compared. The authors first demonstrate that ON-S vs. ON-T RGCs are readily identifiable in a whole mount preparation and respond differently to both static and to a spatially uniform, randomly fluctuating (Gaussian noise) light stimulus. Liner-nonlinear (LN) models were used to estimate the transformation between visual input and excitatory synaptic input for each RGCs; these models suggested the presence of transient versus sustained kinetics already in the excitatory inputs to ON-T and ON-S RGCs. Indeed, the authors show that (glutamatergic) excitatory inputs to ON-S vs. ON-T RGCs are of distinct kinetics. The subtypes of bipolar cells providing input to ON-S are known (i.e., type 6 and 7), but the source of excitatory bipolar inputs to ON-T RGCs needed to be determined. In a tedious process, it is elegantly shown here that ON-T RGCs receive most of their excitatory inputs from type 5 and 6 bipolars. Interestingly, the temporal properties of light-evoked responses of type 5, 6, and 7 bipolars recorded from the somas were indistinguishable and rather sustained, suggesting that the origin of transient kinetics of excitatory inputs to ON-T RGCs suggested by the LN model might be found in the processing of visual signals at the bipolar cell axon terminal. Blocking GABA- or glycinergic inhibitory inputs did not alter the light-evoked excitatory input kinetics to ON-T and ON-S RGCs. Twophoton glutamate sensor imaging revealed significantly faster kinetics of light-evoked glutamate signals at ON-T versus ON-S RGCs. Detailed EM analysis of bipolar cell ribbon synapses onto ON-T and ON-S RGCs revealed fewer ribbon-associated vesicles at ON-T synapses, which is consistent with stronger paired-flash depression of lightevoked excitatory currents in ON-T RGCS versus ON-S RGCs. This study suggests that bipolar subtype-specific differences in the size of synaptic ribbon-associated vesicle pools contribute to transient versus sustained kinetics in RGCs.Strengths:The use of multiple, state-of-the-art tools and approaches to address the kinetics of bipolar to ganglion cell synapse in an identified circuit.Weaknesses:For the most part, the data in the paper support the conclusions, and the authors were careful to try to address questions in multiple ways. Two-photon glutamate sensor imaging experiment showing that blocking GABA- and glycinergic inhibition does not change the kinetics of light-evoked glutamate signals at ON-T RGCs would strengthen the conclusion that bipolar subtype-specific differences in the size of synaptic ribbon-associated vesicle pools contribute to transient versus sustained kinetics in RGCs.

Thank you for this suggestion. We have revised the text throughout to be careful not to imply that amacrine cells have no role in shaping EPSCs and spike output, but instead that the transience of the On-T responses persists without amacrine cells (see for example lines 91, 450-453, 514-518, 696-714). We have also added additional iGluSnFR experiments to the paper to further test this conclusion (new Figure 7). The new data shows that the transience of glutamate release from the On-T cells is retained when (1) spiking amacrine cell activity is suppressed by blocking voltage-gated Na^+^ channels with TTX or (2) all amacrine cell activity is suppressed by blocking AMPA receptors with NBQX. This does provide nice additional evidence that amacrine cells are not necessary for the sustained/transient distinction.

**Reviewer #2 (Public Review):**
Summary:Goal of the study. The authors tried to pinpoint the origins of transient and sustained responses measured at retinal ganglion cells (rgcs), which is the output layer of the retina. Response characteristics of rgcs are used to group them into different types. The diversity of rgc types represents the ability of the retina to transform visual inputs into distinct output channels. They find that the physical dimensions of bipolar cell's synaptic ribbons (specialized release sites/active zones) vary across the different types of cone on-bpcs, in ways that they argue could facilitate transient or sustained release. This diversity of release output is what they argue underlies the differences in on-rgcs response characteristics, and ultimately represents a mechanism for creating parallel cone-driven channels.Strengths:The major strengths of the study are the anatomical approaches employed and the use of the "glutamate sniffer" to assay synaptic glutamate levels. The outline of the study is elegant and reflects the strengths of the authors.Weaknesses:The major weakness is that the ambitious outline is not matched with a complete set of results, and the set of physiological protocols is disjointed, not sufficient to bridge the systems-level question with the presynaptic release question.

Thank you for this comment as it provides an opportunity (here and in the paper) for us to clarify our main goal. We wanted to link the well-established distinction between transient and sustained retinal responses to anatomy. This required locating where this difference arises within the circuitry – which we show to be at least largely the bipolar output synapse – and then examining the structure of this synapse in detail. While we would certainly be interested in connecting our results to a biophysical description of the synapse, that was not the primary focus of our study and was not something we could add without substantial additional work.

Major comments on the results and suggestions.The ribbon model of release has been explored for decades and needs to be further adapted to systems-level work. The study under consideration by Kuo et al. takes on this task. Unfortunately, the experimental design does not permit a level of control over presynaptic/bpc behavior that is comparable to earlier studies, nor do they manipulate release in ways that test the ribbon model (i.e., paired recordings or Ribeye-ko). Furthermore, the data needs additional evaluation, and the presentation and interpretations should draw on published biophysical and molecular studies.

As described above, our goal was to test several possible explanations for the difference between transient and sustained responses in OnT and OnS ganglion cells: (1) differences in the light responses of the bipolar cells that convey photoreceptor signals to the relevant ganglion cells; (2) shaping of bipolar transmitter release by presynaptic inhibition; (3) shaping of ganglion cell responses by postsynaptic inhibition or spike generation; (4) differences in feedforward bipolar synapses. We were surprised to find that the feedforward bipolar synapses play a central role in this difference, and your comment nicely prompts us to relate this to the large literature on biophysical studies of release from ribbon synapses. We have made substantial revisions in the text to do this. This includes anticipating the importance of feedforward synaptic properties in the abstract and introduction (lines 36-37 and 61-64), pointers in the results (lines 539-548), and several new paragraphs in the discussion (starting on lines 751, 773 and 787). By showing that the transient/sustained differences originates largely at feedforward bipolar synapses, we set the stage for future work that shows how biophysical properties of the synapse shape physiological signals that traverse it.

To build a ribbon-centric context, consider recent literature that supports the assertion that ribbons play a role in forming AZ release sites and facilitating exocytosis. Reference Ribeye-ko studies. For example, ribbonless bpcs show an 80% reduction in release (Maxeiner et al EMBO J 2016), the ribbonless retina exhibits signaling deficits at the output layer (Okawa et al ...Rieke, ..Wong Nat Comm 2019), and ribbonless rods show an 80% reduction the readily releasable pool (RRP) of SVs (Grabner Moser, elife 2021). In addition, the authors could refer to whole-cell membrane capacitance studies on mammalian rods, cones, and bpcs, because the size of the RRP of SVs scales with the dimensions and numbers of ribbons (total ribbon footprint). For comparison, bipolars see the review by Wan and Heidelberger 2011. For a comparison of mammalian rods and cones, see, rods: Grabner and Moser (2021 eLife), Mueller.. Regus Leidig et al. (2019; J Neurosci) and cones Grabner ...DeVries (Nat Comm 2023). A comparison of cell types shows that the extent of release is (1) proportional to the total size of the ribbon footprint, and (2) less release is witnessed when ribbons are deleted (also see photo ablation studies by Snellman.... And Mehta..Zenisek, Nat Neurosci and Neuron).

Thank you for these pointers into the literature. We have included much of this work in the revised Discussion (see three paragraphs starting on line 751). The revised text focuses on the evidence that larger and more numerous ribbons lead to increased release. The direct evidence from previous work for this relationship supports our (indirect) conclusions in the current paper about the role of ribbon size and associated vesicle pools in transient vs sustained responses.

Ribbon morphology may change in an activity-dependent manner. The rod ribbon AZ has been reported to lengthen in the dark (Dembla et al 2020), and deletion of the ribbon shortens the length of the AZ (defined by Cav1,4 or RIM2); in addition, the Ribeye-ko AZs fail to change in size with light and dark conditioning. Furthermore, EM studies on rod and cone AZs in light and dark argue that the number of SVs at the base of the ribbon increases in the dark, when PRs are depolarized (see Figure 10, Babai et al 2016 JNeurosci). Lastly, using goldfish Mb1 on-bipolars, Hull et al (2006, J Neurophysio) correlated an increase in release efficiency with an increase in ribbon numbers, which accompanied daylight. >> When release activity is high, ribbon AZ length increases (Dembla, rods), the number of docked SVs increases (Babai, rods cones), and the number of ribbons increases (Hull, diurnal Mb1s).

We have extensively revised the discussion section to include more discussion of ribbons, particularly emphasizing evidence supporting the general argument that larger ribbons support higher release rates. We focused on studies that provided direct links between release rates and ribbon size or number of ribbon-associated vesicles. This includes studies that pair electrophysiology and anatomy and those that measure the consequences of ablating ribbons,

The results under review, Kuo et al., were attained with SBF-SEM, which has the benefit of addressing large-volume questions as required here, yet it achieves lower spatial resolution than what is attained with TEM tomography and FIB-EM. Ideally, the EM description would include SV size, and the density of ribbon-tethered SVs that are docked at the plasma membrane, because this is where the SVs fuse (additional non-ribbon release sites may also exist? Mehta ... Singer 2014 J Neurosci). Studies by Graydon et al 2011 and 2014 (both in J Neurosci), and Jean ... Moser et al 2018 (eLife) are good examples of quantitative estimates of SVs docking sites at ribbons. SBF-SEM does not allow for an assessment of SVs within 5 nm of the PM, but if the authors can identify the number of SVs that appear within the limit of resolution (10 to 15 nm) from the PM, then this data would be useful. Also, what dimension(s) of the large ribbons make them larger? Typically, ribbons are fixed in height (at least in the outer retina, 200 to 250 nm), but their length varies and the number ribbons per terminal varies. Is the larger ribbon size observed in type 6 bpcs do to longer ribbons, or taller ribbons? A longer ribbon likely has more docked SVs. An additional possibility is that more SVs are about the ribbon-PM footprint, either more densely packed and/or expanding laterally (see definitions in Jean....Moser, elife 2018).

We have included an additional analysis of ribbon surface area from our 3D SBFSEM reconstructions. As with the volume measurements included in the original submission, ribbon surface areas are distinct between type 5i and type 6 bipolar cells (Fig. S10A), ON-T RGCs on average receive input from ribbons with smaller surface area than ON-S RGCs (Fig. S10B), and ribbon surface area predicts the number of adjacent vesicles across bipolar cell types (Fig. S10C). We agree that a higher resolution view of presynaptic structures would be very helpful, but the resolution of our SBF-SEM data is limited (e.g. each pixel is 40 nm on a side). This resolution does not allow us to distinguish between vesicles at vs near the membrane.

In our observations, both length and height of the ribbons showed variability across individual bipolar cells. And ribbons in type 6 bipolar cells tended to be either longer and/or taller compared to those in type 5 cells. We agree that a longer ribbon may accommodate more docked SVs. A more definitive analysis would benefit from higher-resolution, isotropic 3D reconstructions of ribbons, which would allow more precise shape analysis and ,together with a detailed assessment of docked SVs at the ribbons.

The ribbon literature given above makes the argument that ribbons increase exocytotic output, and morphological studies suggest that release activity enhances (1) ribbon length (Dembla) and (2) the density of SVs near the PM (Babai). These findings could lead one to propose that type 6 bpcs (inputs to On-sustained) are more active than type 5i (feed into On-transient). Here Kuo et al. show that the bpcs have similar Vm (measured from the soma) in response to light stimulation. Does Vm predict release? Not entirely as the authors acknowledge, because: Cav channel properties, SV availability, and negative feedback are all downstream of bpc Vm. The only experiment performed to test downstream factors focused on negative feedback from amacrines. The data presented in Figures 5C-F led me to conclude the opposite of what the authors concluded. My impression is that the T-ON rgc exhibits strong disinhibition when GABA-blockers are applied (the initial phase is greatly increased in amplitude and broadened with the drug), which contrasts with the S-On rgc responses that show a change in the amplitude of the initial phase but not its width (taus would be nice). Here and in many places the authors refer to changes in release kinetics, without implementing a useful description of kinetics. For instance, take the cumulative current (charge) in Figure 5C and fit the control and drug traces to arrive at taus, and their respective amplitudes, and use these values to describe kinetic phases. One final point, the summary in Figure 5D has a p: 0.06, very close to the cutoff for significance, which begs for more than an n = 5. Given that previous studies have shown that bpc output is shaped by immediate msec GABA feedback, in ways that influence kinetic phases of release (..Mb1 bipolars, see Vigh et al 2005 Neuron), more attention to this matter is needed before the authors rule out feedback inhibition in favor of ribbon size. If by chance, type 5i bpcs are under uniquely strong feedback inhibition, then ribbon size may result from less activity, not less output resulting from smaller ribbons.

The text surrounding Figure 5 led to some confusion, and we have revised that text and the figure for clarity. First, the data in that figure is entirely from On-T cells (the upper and lower panels show block of GABA and glycine receptors separately). Second, the observation that we make there is that block of inhibitory receptors increases the transience of the On-T excitatory input, rather than decreasing it as would be expected if the transience is created by presynaptic inhibition. We have added additional data and that increase in transience is now significant. Inhibitory block does substantially increase the amplitude of the postsynaptic response, and a likely origin of this change in response is inhibitory feedback to the bipolar synaptic terminal. We now indicate this in the text on page 13, lines 438-453.

The key result of this figure for our purposes here is that the transience of the excitatory input to the OffT cell remains with inhibitory input blocked. We have clarified throughout the text that our results indicate that inhibitory feedback is not necessary for the difference between transient release into On-T and sustained release onto On-S. This does not mean that inhibitory feedback does not shape the responses in other ways or contribute to the transient/sustained difference - just that for the specific stimuli we use that difference is retained without presynaptic inhibition. We have also added citations to past work showing that activity of amacrine cells can modulate bipolar transmitter release.

Whether strong feedback inhibition limits activity and therefore limits ribbon size in an activity-dependent way is an intriguing possibility. Indeed, addressing why ribbons are larger in type 6 bipolar cells vs. other bipolar types will be an interesting avenue of further study. However, it would be surprising if ribbon sizes changed during the acute pharmacological block conditions (~10-15 minutes) we employed in our study. Our point here is that there is an interesting correlation between presynaptic ribbon size and the kinetics of glutamate release. We do not think that the two possibilities stated in the last sentence (“…ribbon size may result from less activity, not less output resulting from smaller ribbons”) are mutually exclusive.

We have not further quantified the response kinetics in the experiments of Figure 5 as the large changes induced by the pharmacology (especially GABA receptor block) make it unclear how to interpret quantitative differences. In other places we have quantified kinetics through the STA or specified that our focus was more qualitative (i.e. transient vs sustained kinetics).

As mentioned above, the behavior of Cav channels is important here. This is difficult to address with voltage clamps from the soma, especially in the Vm range relevant to this study. Given that it has previously been modeled that the rod bpc to AII pathway adapts to prolonged depolarization of rbcs through downregulating Cav channel-mediated Ca^2+^ influx (Grimes ....Rieke 2014 Neuron), it seems important for Kou et al to test if there is a difference in Cav regulation between type 6 and 5i bpcs. Ca^2+^ imaging with a GCaMP strategy (Baden....Lagnado Current Biology, 2011) or filling the presynapse with Ca dyes (see inner hair cells: Ozcete and Moser, EMBO J 2020) would allow for the correlation of [Ca]intra with GluSnf signals (both local readouts).

This is a good suggestion but is outside the scope of our current paper. Our focus was on the circuit origin of the difference in response of the OnT and OnS responses rather than the specific biophysical mechanism. We are of course interested in the mechanism, but the additional experiments needed to pin that down would need to be a part of future experiments. The work here represents an important step in that direction as it greatly reduces the number of possible locations and mechanisms for the sustained/transient difference and hence serves to focus any future mechanistic investigations.

Stimulation protocol and presentation of Glutamate Sniffer data in Figure 6. In all of your figures where you state steady st as a % of pk amplitude, please indicate in the figure where you estimate steady state. Alternatively, if you take the cumulative dF/F signal, then you can fit the different kinetic phases. From the appearance of the data, the Sustained Glu signals look like square waves (Figure 6B ROI1-4), without a transient at onset, which is not predicted in your ribbon model that assumes different kinetic phases (1. depletion of docked SVs, and 2. refilling and repriming). The Transient responses (Figure 6B ROI5-8) are transient and more compatible with a depressing ribbon scheme. If you take the cumulative, for all of the On-S and compare it to all of the On-T responses, my guess is the cumulative dF/F will be 10 to 20 larger for the S-On. Would you conclude that bpc inputs to On-S (type 6) release 20fold more SVs per 4 seconds on a per ribbon basis, and does the surface area of the type 6 bpcs account for this difference? From Figures 8B and D, the volume of the ribbon is ~2 fold greater for type 6 vs 5i, but the Surface Area (both faces of ribbon) is more relevant to your model that claims ribbon size is the pivotal factor. If making cumulative traces, and comparisons on an absolute scale is unfounded, then we need to know how to compare different observations. The classic ribbon models always have a conversion factor such as the capacitance of an SV or q size that is used to derive SV numbers from total dCm or Qcontent. See Kim ....et al von Gersdorff, 2023, Cell Reports. Why not use the Gaussian noise stimulus in Fig 6 as in Figure 1 and 2?

For iGluSnFR recordings, steady-state responses were measured from the mean fluorescence over the last 1 sec of the light step (2 sec duration) response. We have included this information in the figure caption and in the Methods.

There is a good deal of variability in the iGluSnR responses from one ROI to another, and the ROIs shown in the original submission had a less prominent transient component than many other ROIs. We have replaced this figure with another that is more representative of the average behavior across ROIs. The full range of behavior is captured in Figure 6C; it is clear across ROIs that glutamate release near ON-S dendrites shows both sustained and transient components. The new experiments in which we block amacrine cell activity also include a few more example ROIs from ON-S cells, and those also show both transient and sustained components.

Your suggestion to integrate the iGluSnFR signals to compare to our structural analysis of ribbons is interesting. However, we are hesitant to make a quantitative comparison between the two without further experiments to validate how the iGluSnFR signals we measure relate to release of single vesicles. For example, a quantitative measure of release based on the iGluSnR experiments would require accounting for possible differences in the expression of the indicator - which could differ both in overall level and/or location relative to release sites.

This comment and one above highlight the importance of measures of ribbon surface area, which we now provide (Figure S10).Figure 7. What is the recovery time for mammalian cones derived from ribbon-based models? There are estimates from membrane capacitance studies. Ground squirrel cones take 0.7 to 1 sec to recover the ultrafast, primed pool of SVs when probed with a paired-pulse protocol (Grabner ...DeVries 2016, Neuron). Their off-bpcs take anywhere from under 0.2 sec to a second to recover, which is a combination of many synaptic factors (Grabner ...DeVries Nat Comm 2023). Rod On bpcs take over a second (Singer Diamond 2006, reviewed Wan and Heidelberger 2011). In Figure 7B, the recovery time is ~150 ms for the responses measured at rgcs. This brief recovery time is incompatible with existing ribbon models of release. Whole-cell membrane capacitance measurements would be helpful here.

Thanks for drawing our attention to this issue. Indeed, we see a relatively rapid recovery in the paired-flash experiments. We now discuss this recovery time in the context of past measurements of recovery of responses in cones and bipolar cells (paragraph starting on line 773). There are many factors that could contribute to the relatively rapid recovery we observe - including synaptic factors such as those highlighted by Grabner et al., (2016) either at the cone-to-bipolar synapses or the bipolar-to-RGC synapses. We are certainly interested in a more detailed understanding of this issue, but the additional experiments are outside the scope of this paper.

Experimental Suggestion: Add GABA blockers and see if type 5i bpc responds with more release (GluSniff) and prolonged [Ca2+] intra (GCaMP). Compare this to type 6 bpc behavior with GABA/gly blockers. This will rule in or out whether feedback inhibition is involved.

Figure 7 in the revised manuscript includes two new experiments examining glutamate release (without the simultaneous measurement of bipolar cell intracellular calcium) while blocking (1) all/most amacrine cell-mediated inhibition via inclusion of NBQX in the bath solution, and (2) blocking spiking amacrine cells via inclusion of TTX in the bath solution. The transient vs sustained difference in light-evoked glutamate release around ON-T and ON-S RGC dendrites remained with amacrine activity suppressed. These new results are consistent with the anatomical and pharmacological data that were included in the initial submission of the manuscript (Fig. 5) that indicate presynaptic inhibition does not have a major role in shaping release kinetics at these synapses.

**Reviewer #3 (Public Review):**
Summary:Different types of retinal ganglion cell (RGC) have different temporal properties - most prominently a distinction between sustained vs. transient responses to contrast. This has been well established in multiple species, including mice. In general, RGCs with dendrites that stratify close to the ganglion cell layer (GCL) are sustained; whereas those that stratify near the middle of the inner plexiform layer (IPL) are transient. This difference in RGC spiking responses aligns with similar differences in excitatory synaptic currents as well as with differences in glutamate release in the respective layers - shown previously and here, with a glutamate sensor (iGluSnFR) expressed in the RGCs of interest. Differences in glutamate release were not explained by differences in the distinct presynaptic bipolar cells' voltage responses, which were quite similar to one another. Rather, the difference in transient vs. sustained responses seems to emerge at the bipolar cell axon terminals in the form of glutamate release. This difference in the temporal pattern of glutamate release was correlated with differences in the size of synaptic ribbons (larger in the bipolar cells with more sustained responses), which also correlated with a greater number of vesicles in the vicinity of the larger ribbons.The main conclusion of the study relates to a correlation (because it is difficult to manipulate ribbon size or vesicle density experimentally): the bipolar cells with increased ribbon size/vesicle number would have a greater possibility of sustained release, which would be reflected in the postsynaptic RGC synaptic currents and RGC firing rates. This model proposes a mechanism for temporal channels that is independent of synaptic inhibition. Indeed, some experiments in the paper suggest that inhibition cannot explain the transient nature of glutamate release onto one of the RGC types. Still, it is surprising that such a diverse set of inhibitory interneurons in the retina would not play some role in diversifying the temporal properties of RGC responses.Strengths:(1) The study uses a systematic approach to evaluating temporal properties of retinal ganglion cell (RGC) spiking outputs, excitatory synaptic inputs, presynaptic voltage responses, and presynaptic glutamate release. The combination of these experiments demonstrates an important step in the conversion from voltage to glutamate release in shaping response dynamics in RGCs.(2) The study uses a combination of electrophysiology, two-photon imaging, and scanning block-face EM to build a quantitative and coherent story about specific retinal circuits and their functional properties.Weaknesses:(1) There were some interesting aspects of the study that were not completely resolved, and resolving some of these issues may go beyond the current study. For example, it was interesting that different extracellular media (Ames medium vs. ACSF) generated different degrees of transient vs. sustained responses in RGCs, but it was unclear how these media might have impacted ion channels at different levels of the circuit that could explain the effects on temporal tuning.

We do not have an explanation for the quantitative differences in response kinetics we observed in Ames’ medium vs. ACSF. There are modest differences in calcium and magnesium concentration and a larger difference in potassium (2.5 mM in ACSF vs 3.6 mM in Ames). It would be interesting to test which of these (or other) differences accounts for the difference in response kinetics.

(2) It was surprising that inhibition played such a small role in generating temporal tuning. At the same time, there were some gaps in the investigation of inhibition (e.g., IPSCs were not measured in either of the RGC types; pharmacology was used to investigate responses only in the transient RGCs).

We were also surprised at this result. We have included additional data on inhibition in the revised manuscript. Figure S3 shows light-evoked IPSC data from both RGC types (Fig. S3) and Fig. 7 shows additional iGluSnFR measurements around both ON-T and ON-S RGC dendrites with inhibition blocked via bath application of NBQX (Fig. 7) and separately with inhibition from spiking amacrine cells blocked with TTX. These experiments provide additional evidence for the small role of inhibition. We attempted to measure the kinetics of excitatory input to ON-S cells with inhibition blocked, but we found that the excitatory input showed strong spontaneous oscillations under these conditions and the light responses were changed so drastically that we did not feel we could make a clear comparison with control conditions.

(3) There could be additional discussion and references to the literature describing several topics, including: temporal dynamics of glutamate release at different levels of the IPL; previous evidence that release sites from a single presynaptic neuron can differ in their temporal properties depending on the postsynaptic target; previous investigations of the role of inhibition in temporal tuning within retinal circuitry.

Thanks, we have included more discussion and references to the relevant literature as you have suggested in the recommendations to authors.

**Reviewer #1 (Recommendations For The Authors):**
The presented raw data of the pharmacological experiments show that SR95531 and TPMPA robustly increased both the amplitude and duration of the transient component of the light step-evoked excitatory currents, with slight, if any enhancement of the sustained component in ON-T RGCs Figure 5C. Statistical analysis of the population data (n=5) with Wilcoxon signed rank test yielded no significant difference (ln 363). However, reanalyzing the data extracted from the graph (Figure 5D) revealed that the difference between the paired observations is normally distributed (Shapiro-Wilk normality test, P=0.48) allowing parametric statistics to be used, which provides higher statistical power. Accordingly, reanalyzing the presented data with paired Student's t-test data revealed significant differences (P=0.01) in the steady-state amplitude normalized to that of the peak, recorded in the presence of SR95531 and TPMPA. In other words, based on the (rough) analysis of the presented pharmacology data GABAergic feedback inhibition significantly contributes to shaping the transient portion of the light-evoked excitatory currents in ON-T RGCs, by making it more transient. I believe a similar analysis based on the actual data is necessary, and the results should be communicated either way. However, if warranted, two-photon glutamate sensor imaging experiments showing that blocking GABA- and glycinergic inhibition does not change the kinetics of light-evoked glutamate signals at ON-T RGCs should also be performed, as these would be critical in drawing a conclusion regarding the effect of feedback inhibition on glutamate release from bipolar cells.

Thanks for this feedback. We have added another cell to the data set in Fig. 5D. With this addition, SR95531/TPMPA application significantly increases the response transience of excitatory currents measured in ON-T RGCs compared to control. This enhanced transience in GABA_A/C_ receptor blockers is due to an increase in the amplitude of the initial peak component of the response (control peak amplitude: -833.7±103.3 pA; SR95531+TPMPA peak amplitude: 2023±372.7pA; p=0.03, Wilcoxon signed rank test), with no change to the later sustained component (control plateau amplitude: -200.7±14.71pA; SR95531+TPMPA plateau amplitude: -290.9±43.69pA; p=0.15, Wilcoxon signed rank test).

We should clarify that this result indicates that GABAergic inhibition makes the excitatory inputs to ON-T RGCs less transient. Block of GABA receptors increased transience, thus intact GABAergic transmission appears to limit the initial peak of the response and therefore make excitatory currents more sustained. We unfortunately were not able to examine whether sustained excitatory currents in ON-S RGCs would become more transient using the same approach. In our hands, bath application of SR95531+TPMPA led to the generation of large-amplitude (>1nA) oscillatory bursts of excitatory input that developed within 5 minutes and persisted for the duration of the incubation (up to ~30 min) in drugs. Further, presentation of light steps tended to induce variable amplitude responses, likely dependent on the presence of spontaneous bursts; when large amplitude responses were evoked, these typically oscillated for several seconds after the step.

To examine a potential role for presynaptic inhibition in transient vs. sustained bipolar cell output, we therefore chose to eliminate amacrine cell-mediated inhibition by bath application of the AMPA/kainate receptor antagonist NBQX in additional iGluSnFR measurements. This manipulation should leave ON bipolar cell responses intact while eliminating most amacrine cell-mediated responses (and OFF bipolar cell driven responses). In separate experiments, we also eliminated inhibition from spiking amacrine cells by bath application of TTX. As shown in new Fig. 7, sustained and transient responses persisted in distal versus proximal RGC dendrites, respectively. Compared to SR95531/TPMPA, bath application of NBQX was not associated with spontaneous bursts of glutamate release around ON-S dendrites. These results show that amacrine cell-mediated inhibition is not required for either sustained or transient glutamate release from bipolar cells that provide input to ON-S and ON-T RGCs.

Small points:(1) The legend of Figure 1 (D) refers to shaded areas to show {plus minus} SEM, but no shade is visible (at least in my printout).

The SEM shading is there in Fig. 1D but is mostly obscured by the mean lines for the respective RGC types. We have added this to the figure caption.

(2) I found the reported Vrest for the ON bipolar cells somewhat depolarized. Perhaps due to the uncompensated junction potentials?

These measurements are indeed not corrected for the liquid junction potential (which is approximately -10.8 mV between K-gluconate internal and Ames’ solution). We did not apply this correction since the appropriate value is not clear in perforated patch recordings as the intracellular chloride concentration is unknown (and can differ from that in the pipette solution). We have clarified this in the results text where we describe the Vrest values (lines 335-338).

(3) It is Wilcoxon signed rank test, not Wilcoxan.

Thanks for catching this. This has been corrected in the revised manuscript.

**Reviewer #2 (Recommendations For The Authors):**
Some amacrines express vesicular Glut-3 transporter and are reported to release glutamate (Marshak, Vis Neurosci 2016). Are Amacrine vGlut3 signals postsynaptic (within ~0.5 um) to cone bpc ribbons?

We did not characterize VgluT3-expressing amacrine cells in our SEM datasets. A recent study by Friedrichson et al. (Nat. Comm. 2024; PMID 38580652) using 3D SEM reconstructions found that Vglut3-amacrines are postsynaptic to both type 5i and type 6 bipolar cells, as well as other type 5/xbc bipolar cells (and receive >50% of their input from type 3a OFF bipolar cells).

How far apart are the postsynaptic targets from the ribbon release sites? The ribbons at type 5i bpc/On-T input appear separated from the dendrites of On-T rgcs (Figure 8C). At least further away than the type 6 bpc ribbons are from On-S rgc dendrites (Figure 8C). Distance may create a thresholding phenomenon, whereby only multivesicular bouts at the onset of depolarization are able to elevate synaptic Glu to levels needed to activate On-T GluRs. See Grabner et al Nat Comm 2023 for such scenarios in the outer retina.

This is an intriguing possibility, but we should point out that the presynaptic ribbons in Fig. 9C (former Fig. 8C) are similar distances (within the resolution of our reconstructions) from the ON-T and ON-S dendrites. We have increased the brightness of the dendrite segments for both RGC types in the resubmission figure; note that ON-T RGCs have spine-like protrusions that may not have been as apparent in the previously submitted version of our manuscript.

In Figures 1 and 2, Sustained responses look like the derivative of Transient responses, minus the negative going inflection. In addition, the sustained responses appear to have a lower threshold of activation than the transient On rgcs, because there are more bouts of action potentials (and membrane depol in V-clamp) with earlier onset in sustained than transients traces. It would be great if the GLuSniff data captured these differences. Take cumulative dF/F and see what the onset time is, or an initial tau if possible.

This is a good suggestion. However, we are reluctant to make detailed quantitative comparisons such as this without further validation of how the kinetics of the iGluSnFR signals relate to kinetics of glutamate release. A specific concern is that differences in the location and amount of iGluSnFR expression could impact any such comparisons.

A recent study by Kim et al von Gersdorff (Cell Reports, 2023) presents interesting phases of release in response to light flashes, measured from AIIs, and complementary results from pairs of rbcs-AIIs. The findings highlight the complexity of SV pools under well-controlled experiments. Could their results be explained as variations in rbc ribbon size through development, and possibly between rbcs or within an rbc?

This certainly seems possible and would be consistent with the dependence of release on ribbon size that our results support. It would be interesting to see if there are clear anatomical correlates of that change in release properties.

Figure 5 is a pivotal point in the study, but my review has identified numerous weaknesses. The feedback inhibition onto bipolar cell terminals is likely to sculpt glutamate release, and the results do not convincingly rule out this possibility. The suggestions for improvements range from the data needing to be reanalyzed with regard to statistical tests, and/or adding a few more data points (n = 5) before concluding a p: 0.06 is insignificant.

We have added an additional recording to this data set. With n = 6 cells, there is now a statistically significant difference between ON-T RGC excitatory currents measured in control conditions versus during GABA_A/C_ receptor blockade. Please note that all the recordings shown in Figure 5C-F are from ON-T RGCs (the two panels show separately block of GABergic and glycinergic receptors). We did not make it sufficiently clear that the original trend (now statistically significant) is opposite of that expected if presynaptic GABAergic inhibition contributes to response transience in ON-T RGCs. What we see is that excitatory synaptic inputs to ON-T RGCs become more transient (rather than mpre sustained) during GABA_A/C_ receptor blockade. We have revised the text in that section to make this point more clearly.

We have also included new data from iGluSnFR measurements showing that bath application of NBQX does not affect light step-evoked glutamate release kinetics at proximal (sustained) or distal (transient) RGC dendrites (control: steady-state amp. as % of peak amp. 13 ± 10; mean ± S.D.; n = 189 ROIs/4 FOVs for ON-T dendrites vs 40 ± 12; mean ± S.D.; n = 287 ROIs/8 FOVs for ON-S dendrites; NBQX: 6 ± 3; mean ± S.D.; n = 112 ROIs/1 FOV for ON-T dendrites vs 23 ± 9; mean ± S.D.; n = 97 ROIs/2 FOVs for ON-S dendrites; *p<0.001). By blocking glutamate receptors on amacrine cells, NBQX (AMPA/KAR antagonist) eliminates all/most amacrine cell-mediated signaling in the retina and should therefore abolish presynaptic inhibitory input to bipolar cell terminals across the IPL. Taken together, our results indicate that presynaptic inhibition does not play a critical role in establishing transient versus sustained kinetics for the stimulus conditions we employed in our study.

There is a need to cite more recent literature on bipolar cell ribbons (e.g. see Wakeham et al., Front. Cell. Neurosci., 2023), in order to support experimental design and interpretation of the results. The authors should discuss their Ribeye-KO data from Okawa et al 2019 Nat Comm, Figure 7, in the context of their new iGluSnFR results.

Thank you for prompting us on this issue. We have expanded the discussion regarding ribbons and included more citations to the ribbon literature. That is largely in the three paragraphs starting on line 727.

One point deserves emphasis because it is central to the authors' ribbon model but not consistent with their data. The ribbon model as they put it, and as commonly stated, holds that a transient phase of release at the onset of depolarization indicates the depletion of the primed SVs, and the subsequent slower rate of release (steady state release in the authors' terms) reflects recruiting, priming, and release of new SVs. The On-transient dendrite GluSnf responses agree with this multiphasic process, but the sustained responses show only an elevation in glutamate without a pronounced initial peak, creating a square-wave-shaped response (Figure 6B). This does not agree with the simple ribbon-based release model. I would expect the signals from the T- and S-on dendrites to have a comparable initial phase, while the sustained phase should be greater in amplitude for the S-on dendrites. More discussion may clarify possible mechanisms.

Thanks for pointing this out. The example iGluSnFR traces we originally included in the manuscript were not entirely representative in that they did not show much initial transient phase. Note there is a distribution of steady-state amplitudes for proximal dendrites in Fig. 6C; the examples are from ROIs from the upper end of the distribution. In the new Figure 7, we have included some additional examples that show both a clear transient and sustained component. The summary data in Figure 6C shows the distribution of sustained/transient ratios across ROIs.

**Reviewer #3 (Recommendations For The Authors):**
(1) It would be interesting to understand the differences in IPSCs in the two RGC types. Perhaps they are small in both types, which would explain their apparent lack of impact on temporal tuning. The authors may already have these data.

We did make measurements of noise-evoked IPSCs (as well as EPSCs) in a subset of ON-T and ON-S recordings. We have now included this data as Figure S3. There are slight differences in the kinetics of inhibition between RGC types (Fig. S3C) and there is a trend towards stronger inhibition (relative to excitation) in ON-T RGCs compared to ON-S RGCs (Fig. S3E), although there is not a statistically significant difference. In both cases excitatory synaptic currents are as large or larger than inhibitory currents, and this does not include the difference in driving force near spike threshold which will favor excitatory input by a factor of 2-3. Hence our data suggests that postsynaptic inhibition does not play a major role in generating the differential temporal spiking responses of ON-T and ON-S RGCs. However, additional experiments examining the relative contribution of excitation and inhibition to spiking output in these RGCs would be needed to reach a firm conclusion.

The pharmacological experiments in which we blocked inhibition (Fig. 5C-F, new Fig. 7) were designed to test the effect of presynaptic inhibition on bipolar cell output (voltage-clamp isolation of excitatory currents in Fig. 5; iGluSnFR measurements of glutamate release in Fig. 7). We do not mean to suggest that postsynaptic inhibition does not have any role in shaping the spiking behavior of these RGC types, but that transient vs. sustained kinetics are already present in the bipolar cell output and that presynaptic inhibition of bipolar cell terminals does not appear to account for this difference. We have revised the text throughout to be clearer on this point.

(2) It could be convincing to show transient/sustained differences between RGC types in dim light, where the response would depend on the rod bipolar/AII circuit. In this case, any difference in temporal properties would presumably be explained by differences that localize to the cone bipolar cell axon terminals. Indeed, is that the result in Figure 1B? This seems to be a dim stimulus presented on darkness, which may be driven through the rod bipolar pathway. The authors could then discuss the interpretation of this data in terms of the rod bipolar circuit.

Yes, Figure 1B is a dim light step (~30R*/rod/s) presented from darkness and the distinction between cells is clear down at still lower light levels that more effectively isolate signaling through the rod bipolar pathway. Thanks for making this point that observation of distinct temporal responses under scotopic conditions where signals suggests these differences must arise at and/or downstream of cone bipolar cell output. We have included additional text (lines 361-365) in the results describing bipolar cell responses that raise this point.

(3) Glutamate release was already measured across the full IPL depth by Borghuis et al. (2013) and Franke et al. (2017). It would be appropriate to better motivate the current study based on these existing measurements.

We have clarified that these important studies provided important motivation for measuring excitatory synaptic input to ON-T vs. ON-S RGCs (lines 165-169).

(4) Line 212/213. It would be appropriate to add to the list of papers showing the different stratification of transient vs. sustained responses: Borghuis et al. (2013) and Beaudoin et al. (2019).

Thank you - these references have been added.

(5) Line 635-638. It would be useful to discuss papers by Pottackal et al. (2020, 2021), which suggested that a single presynaptic cell (starburst) can signal with different temporal properties depending on the postsynaptic target (other starburst vs. DSGCs). The mechanism was not completely resolved (i.e., it was not explained by differences in presynaptic Ca channels at the two synapse types), but it at least shows that neurotransmitter release can show different filtering depending on the postsynaptic target from the same presynaptic neuron. (This could also be at play for the type 6 bipolar cell inputs to ON-S vs. ON-T RGCs in the present study.)

We have added a reference to Pottackal et al 2021 in this section.

(6) Line 714. Should describe the procedure for embedding the tissue in agarose.

We have added more detail regarding agarose embedding for preparation of retinal slices in the methods.

(7) Line 775. Need a better description of the virus (not the construct), what serotype? Provide the Addgene number if available.

This has been added to the methods.

(8) Line 808. Was the SD for the gaussian really 50%? That would cut off a lot of the distribution, i.e., it would get clipped at 0.

Yes, the SD for Gaussian noise was 50%. This high contrast stimulus was used in part to achieve measurable signals from bipolar cells. You are correct that some of the distribution was clipped at 0 (it was also clipped at twice the mean to make sure that the distribution remained symmetrical). The clipping was accounted for during our LN analyses.

(9) The paper should discuss Swygart et al. (2024) results showing different spatial surround properties of neighboring synapses from a type 6 bipolar cell. Based on this result, it would seem very likely that amacrine cells could play a role in shaping the temporal processing of bipolar cell glutamate release as well. Indeed, spatial and temporal processing will not be completely independent in a typical experiment. For example, with the spot stimulus used in the present study, bipolar cells within the center versus the edge of the spot will have different balances of center/surround activation, which could potentially influence their temporal processing.

We have included discussion of results from Swygart et al 2024 in the section of the Discussion in which we point out differences in surround inhibition between ON-S and ON-T RGCs (lines 710-714). We agree that spatial and temporal processing are not completely independent. Our results with SR95531/TPMPA indicate ON-T RGCs receive stronger GABAergic surround inhibition than ON-S RGCs (Fig. S8). However, our results in Fig. 5C-D show GABAergic surround inhibition makes ON-T excitation more sustained rather than more transient. So even though bipolar cells presynaptic to ON-T RGCs receive stronger surround inhibition (Fig. S8), this inhibition does not establish the transient kinetics of glutamate release from these bipolar cells (in fact, it works to make release more sustained). Additional iGluSnFR experiments where we used NBQX to block all/most amacrine cell-mediated responses also suggest presynaptic inhibition does not have an important role in establishing differential glutamate release kinetics onto ON-S vs. ON-T RGC dendrites (Fig. 7).

(10) Cui et al. 2016 described ON-S Alpha as having a divisive suppression mechanism that explained the temporal properties of white-noise response better than a standard LN model. Do the authors think the divisive suppression reflects a property of the excitatory synapses independent of inhibition?

This is an interesting question, but one for which we don’t have a good answer for now. As mentioned in some of the above responses and as we have tried to clarify in the manuscript, we do not mean to imply that there is no role for presynaptic inhibition in modulating bipolar cell output, including for the divisive suppression described by Cui et al. Rather, our point is that the distinction between transient and sustained excitatory input to ON-T and ON-S RGCs does not require presynaptic inhibition and is more likely an intrinsic property of the bipolar cell synapses.

(11) Do the authors mean to imply that the pool size at bipolar cell ribbon synapses could depend on the use of Ames vs. ACSF?

For now, we do not have a good answer as to why there are quantitative differences in response kinetics between Ames and ACSF. We have not done any experiments to investigate whether ribbon sizes or ribbon pools are different in the different solutions.

(12) More generally, different mean luminance levels could drive different levels of baseline glutamate release, which could alter the available pool of vesicles at bipolar cell ribbon synapses. Can we explain varying degrees of transient/sustained in the same cell at different levels of mean luminance based on this mechanism (e.g., Grimes et al., 2014)?

Yes, the emergence of a transient component of excitatory input to ON-S RGCs at ~100 R*/rod/s versus at scotopic levels (0.5 R*/rod/s) in Grimes et al. (2014) could be due to differences in the number of releasable vesicles (due to different type 6 bipolar cell axon terminal membrane potentials and hence differences in spontaneous release rates) at the different light levels.

We should note that although ON-T and ON-S RGCs exhibit some changes in transient/sustained kinetics across different light levels, the relative differences between these RGC types are preserved across light levels. We have included a statement about this in the text (lines 361-367).

(13) Figure 1. Have the authors considered performing the LN analysis of the firing responses, to compare the degree of rectification between the two RGC types?

This is a good suggestions. From an LN analysis of spiking responses, we do not observe a clear difference between the static nonlinearity component of the model for ON-T and ON-S RGCs. Both RGC types are strongly rectified under our experimental conditions.

(14) Figure 5. Do the authors have the pharmacology data for the ON-S cells? There are examples of sustained EPSCs in amacrine cells that become more transient after blocking inhibition, which at least suggests that inhibition can play some role in the transient/sustained nature of glutamate release (Park et al., 2015, Figure 3). Perhaps ON-S cells likewise become more transient with inhibition blocked.(The colored symbols in A were not visible in a printout. It would be useful to indicate the cell type (ON-T) in C and E).

As described above in the response to reviewer 1’s recommendation for authors, we were not able to use SR95531/TPMPA for recordings from ON-S RGCs. Bath application of these drugs led to oscillatory bursts of excitatory input to ON-S RGCs. However, the lack of effect of bath-applied NBQX on the kinetics of glutamate release around either ON-T or ON-S RGC dendrites (new Fig. 7) suggests that presynaptic inhibition does not contribute to generating sustained excitation to ON-S RGCs (or transient excitation to ON-T RGCs).

We have corrected Fig. 5A to include the referenced colored symbols and have also edited Fig 5C and E to clarify that measurements in Fig. 5C-F are from ON-T RGCs.

(15) Figure 6 legend. Should be Kcng4-Cre, not KCNG-Cre. Also, it should make clear that this is cre-dependent expression of iGluSnFR. For C, were the statistics based on the number of FOVs?

Thanks for catching this, we have corrected Figure 6 legend. The methods section includes a description of how we achieved iGluSnFR expression on alpha RGC dendrites via a cre-dependent viral strategy in Kcng4-Cre mice. We have also clarified that the statistics are based on ROIs in Figure 6C.

(16) Figure 7, Flashes were apparently 400% contrast on a dim background. What was the background? Is there a rod component to the response in this case?

In Figure 7 (now Figure 8), the same background (~3300 R*/rod/s; 2000 P*/Scone/s) was used as in the Gaussian noise and step response experiments. At this light level, the response should be primarily be mediated by cones.

(17) Figure S1. The colors here differ from those in previous figures (Here, ON-T, magenta; ON-S, cyan). Is something mislabeled?

Thanks for catching this. We mistakenly swapped the labels in the legend for Fig. S1. The figure colors were correct, but we have corrected the legend in the revised manuscript.

(18) Figure S2. For the LN model for RGC synaptic currents, the ON-S are more rectified than some previous recordings (Cui et al., 2016). Is this perhaps explained by different light levels?

We aren’t sure why ON-S excitatory currents are more strongly rectified in our recordings compared to Cui et al., 2016. Cui et al. used an ~20-fold higher background light intensity (~40,000 P*/cone/s vs. ~2000 P*/cone/s in our study), so different light levels may be a factor although we should point out that rectification increases in these RGCs between scotopic to low photopic light levels (see Grimes et al., 2014 and Kuo et al., 2016).

(19) The study is apparently comparing PV1 and PV2 described in Farrow et al. (2013; see Supplementary information for stratification analysis), which should be cited.

Thanks, we have corrected this oversight in the revised manuscript. We now cite Farrow et al and mention the connection to PV1 and PV2 in the first paragraph of Results (lines 104-108).